# Alterations in the intrinsic properties of striatal cholinergic interneurons after dopamine lesion and chronic L-DOPA

Se Joon Choi[1], Thong C Ma[2], Yunmin Ding[2], Timothy Cheung[2], Neal Joshi[2], David Sulzer[1], Eugene V Mosharov[1]*, Un Jung Kang[2]*

[1]Department of Psychiatry, Columbia University Medical Center, New York, United States; [2]Department of Neurology, Grossman School of Medicine, New York University, New York, United States

**Abstract** Changes in striatal cholinergic interneuron (ChI) activity are thought to contribute to Parkinson's disease pathophysiology and dyskinesia from chronic L-3,4-dihydroxyphenylalanine (L-DOPA) treatment, but the physiological basis of these changes is unknown. We find that dopamine lesion decreases the spontaneous firing rate of ChIs, whereas chronic treatment with L-DOPA of lesioned mice increases baseline ChI firing rates to levels beyond normal activity. The effect of dopamine loss on ChIs was due to decreased currents of both hyperpolarization-activated cyclic nucleotide-gated (HCN) and small conductance calcium-activated potassium (SK) channels. L-DOPA reinstatement of dopamine normalized HCN activity, but SK current remained depressed. Pharmacological blockade of HCN and SK activities mimicked changes in firing, confirming that these channels are responsible for the molecular adaptation of ChIs to dopamine loss and chronic L-DOPA treatment. These findings suggest that targeting ChIs with channel-specific modulators may provide therapeutic approaches for alleviating L-DOPA-induced dyskinesia in PD patients.

*For correspondence:
em706@cumc.columbia.edu (EVM);
un.kang@nyulangone.org (UJK)

Competing interests: The authors declare that no competing interests exist.

## Introduction

The loss of dopamine (DA) neurons of the substantia nigra pars compacta results in depletion of striatal DA and the manifestation of the cardinal symptoms of Parkinson's disease (PD). Though DA replacement therapy with its metabolic precursor L-3,4-dihydroxyphenylalanine (L-DOPA) remains the most effective treatment for PD symptoms, L-DOPA therapy can cause debilitating involuntary movements termed L-DOPA-induced dyskinesias (LID) (*Fahn, 2015*). Dysfunction in striatal neurocircuitry due to nigrostriatal neurodegeneration and non-physiological DA synthesis from L-DOPA by non-dopaminergic neurons are thought to underlie the development of LID. The striatum is comprised primarily of GABAergic spiny projection neurons with a small population of GABAergic and cholinergic interneurons that in mice constitute ~5% of all striatal neurons. Though small in number, these interneurons have widespread effects on basal ganglia function (*Abudukeyoumu et al., 2019*; *Goldberg et al., 2012*) and are thought to be important contributors to both PD-related motor dysfunction and LID.

In particular, striatal cholinergic interneurons (ChIs) have been shown to contribute to movement abnormalities and LID development in rodent models of PD (*Aldrin-Kirk et al., 2018*; *Bordia et al., 2016b*; *Ding et al., 2011*; *Divito et al., 2015*; *Gangarossa et al., 2016*; *Lim et al., 2015*; *Shen and Wu, 2015*; *Won et al., 2014*). Using mice deficient for paired-like homeodomain transcription factor 3 (Pitx3) which lack substantia nigra DA neurons from birth, we previously reported that repeated L-DOPA treatment caused LID, increased baseline ChI firing rate, and potentiated ChI response to DA (*Ding et al., 2011*), while ablation of ChIs in a 6-hydroxydopamine (6-OHDA) model significantly attenuated LID (*Won et al., 2014*). Similarly, increasing ChI firing rate by chemogenetic or

optogenetic methods enhanced LID expression (*Aldrin-Kirk et al., 2018*; *Bordia et al., 2016a*), while decreasing ChI firing rate by inhibiting H2 histamine receptors that are preferentially expressed in ChIs in the striatum, attenuated LID (*Lim et al., 2015*). Alterations in ChI activity following DA depletion have been studied, although with conflicting results (*Ding et al., 2006*; *Maurice et al., 2015*; *McKinley et al., 2019*; *Sanchez et al., 2011*; *Tubert et al., 2016*). However, the basis of changes in ChI physiology resulting from chronic L-DOPA treatment of DA depleted mice has not been identified.

Here, using acute striatal slice electrophysiological recordings in the dorsolateral striatum prepared from DA depleted and L-DOPA treated mice, we show that changes in intrinsic properties of ChIs cause slower spontaneous firing rates following DA depletion and faster firing rates after chronic L-DOPA treatment compared to sham controls. We found that both HCN and SK current were decreased in the DA depleted condition, whereas only SK current remained depressed after chronic L-DOPA. Changes in the activities of these two channels were sufficient to explain the observed alterations in ChI firing. Targeting altered activity of striatal ChIs with specific channel modulators may provide a potential therapeutic approach for the alleviation of LID in PD patients.

## Results

### Changes in ChI spontaneous firing rate following DA depletion and chronic L-DOPA treatment

To study alterations in ChI physiology in the parkinsonian mouse striatum, we induced dopaminergic lesions by infusing 6-OHDA unilaterally to the medial forebrain bundle (MFB). The protocol that we used produces severe and permanent lesion of both the cell bodies and axons of midbrain dopaminergic neurons (*Won et al., 2014*). After 3–4 weeks of recovery, the animals were randomized to receive daily IP injections of either saline (*6-OHDA* group) or L-DOPA (3 mg/kg, *chronic-LD* group) that continued for the next 3–11 weeks. Control mice (*sham* group) received vehicle MFB infusion and daily IP saline injections (*Figure 1A*). All 6-OHDA-infused mice showed severe deficits in contralateral front paw adjusting steps in the weight-supported treadmill stepping task, confirming significant lesion of the dopaminergic system. There was no difference in stepping deficit between mice assigned to the 6-OHDA or chronic-LD groups (% steps taken with impaired paw; 6-OHDA: 9.6 ± 0.9%, n = 21 mice; chronic-LD: 8.8 ± 0.7%, n = 30 mice; p=0.547, Mann-Whitney test; *Figure 1—figure supplement 1A*). When challenged with a dyskinesogenic dose of L-DOPA (3 mg/kg), all lesioned mice showed abnormal involuntary movements indicative of LID, starting with the first L-DOPA injection (*Figure 1—figure supplement 1B*). Total LID magnitudes were similar between the first L-DOPA dose (representative of the 6-OHDA group) compared to mice tested after chronic L-DOPA (total LID score; first dose: 45.1 ± 2.7, n = 11 mice; chronic-LD: 42.3 ± 3.4, n = 23 mice; p=0.295, Mann-Whitney test; *Figure 1—figure supplement 1C*). However, the onset of LID was shifted to earlier time points in chronic-LD mice (LID score for the first 10 min; first dose: 6.0 ± 0.7; chronic-LD: 13.4 ± 1.2; p=0.0002, Mann-Whitney test; *Figure 1—figure supplement 1B,D*), indicating further sensitization of LID despite maximized total LID scores from the dose of L-DOPA we used (*Nadjar et al., 2009*).

Next, using cell-attached recordings we measured spontaneous action potentials (sAP) of ChIs in acute striatal slices from the three groups of mice. Cholinergic neurons were identified by their large soma size relative to other cells in the dorsolateral striatum. The spontaneous firing frequency of ChIs decreased in 6-OHDA lesioned mice, but this decrease was reversed by chronic L-DOPA treatment to a level higher than that of sham-lesioned mice (sham: 2.6 ± 0.2 Hz, n = 83 neurons/26 mice; 6-OHDA: 1.7 ± 0.2 Hz, n = 81 neurons/29 mice; chronic-LD: 3.5 ± 0.2 Hz, n = 87 neurons/25 mice; p<0.0001, Kruskal-Wallis test; pairwise: sham vs. 6-OHDA, p=0.0002; sham vs. chronic-LD, p=0.0029; 6-OHDA vs. chronic-LD, p<0.0001, Dunn's multiple comparison test; *Figure 1B,C*). These changes persisted in the presence of blockers of ionotropic GABA and glutamate receptors (*Figure 1C*), indicating that altered spontaneous firing was driven by ChI-intrinsic mechanisms.

ChIs exhibit cell autonomous firing patterns, including irregular firing with bursts and pauses (*Goldberg and Wilson, 2005*; *Wilson, 2005*; *Wilson and Goldberg, 2006*). Previous reports showed that DA depletion decreased the regularity of spontaneous firing and the number of bursts and pauses (*McKinley et al., 2019*). Consistent with these findings, we found an increase in the

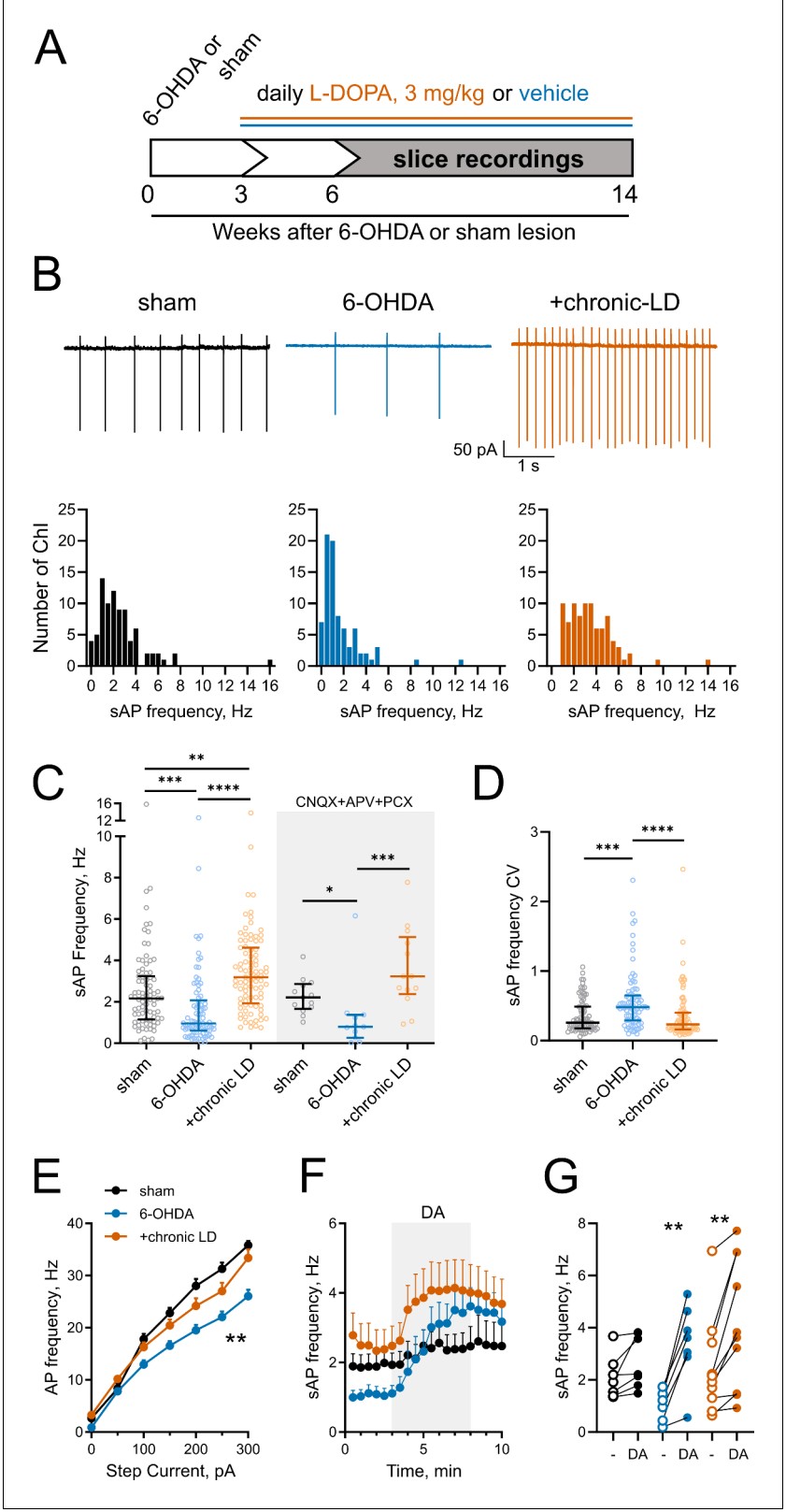

**Figure 1.** Changes in ChI spontaneous firing frequency induced by DA depletion followed by chronic L-DOPA treatment. (**A**) DA lesion and chronic L-DOPA treatment paradigm. 3–4 weeks after unilateral 6-OHDA lesion, mice were randomly divided into two groups to receive either saline or L-DOPA. Experimental groups included sham: mice with vehicle injection into the MFB, 6-OHDA: mice with MFB lesions injected with daily IP saline, and chronic
*Figure 1 continued on next page*

*Figure 1 continued*

LD: MFB-lesioned mice treated with 3 mg/kg L-DOPA IP once daily. Electrophysiological slice recordings were carried out 3–11 weeks after the initiation of L-DOPA or saline injections. (**B**) Representative cell-attached recordings and distributions of average (per cell) instantaneous spontaneous action potential frequency (sAP) of ChIs from sham-lesioned (n = 83 neurons/26 mice), 6-OHDA-lesioned (n = 81 neurons/29 mice), and 6-OHDA-lesioned mice treated with chronic LD (n = 87 neurons/25 mice). Scale bars are 1 s and 50 pA. (**C**) Dot plots of spontaneous cell activity in the absence (same as in B) and the presence of synaptic blockers CNQX (10 µM), APV (25 µM) and picrotoxin (PCX) (25 µM). Number of recordings with synaptic blockers: sham n = 13 neurons/3 mice, 6-OHDA n = 11 neurons/3 mice, chronic-LD 13 neurons/3 mice. (**D**) Coefficient of variation (CV) of instantaneous sAP frequencies in sham, 6-OHDA, and chronic LD groups (same N as in B). For C and D, line denotes median, error bars show interquartile range, p<0.05 (*), p<0.01 (**), p<0.001 (***), or p<0.0001 (****) by Kruskal-Wallis test with Dunn's multiple comparison test; (**E**) The number of evoked action potentials following current injection was decreased in ChIs from 6-OHDA lesioned mice but restored to sham levels after chronic L-DOPA treatment. p<0.01 (**), 6-OHDA vs. the two other groups by two-way ANOVA with Tukey's post-hoc test; sham n = 11 neurons/4 mice, 6-OHDA n = 16 neurons/5 mice, chronic-LD n = 18 neurons/5 mice. (**F**) Averaged perforated-patch recordings of sAP in ChIs following 30 µM DA perfusion in the presence of synaptic blockers. Sham n = 7 neurons/3 mice, 6-OHDA n = 7 neurons/2 mice, chronic-LD n = 10 neurons/3 mice. (**G**) Changes in average sAP frequencies in individual cells before and after DA exposure. (same N as in F). p<0.01 (**) by paired t-test.

The online version of this article includes the following source data and figure supplement(s) for figure 1:

**Source data 1.** Individual neuron data and statistics for all panels and figure supplements.
**Figure supplement 1.** Expression of L-DOPA-induced dyskinesia (LID) in 6-OHDA-lesioned mice.
**Figure supplement 2.** Burst-pause activity in ChIs.
**Figure supplement 3.** Basic electrophysiological characteristics in ChIs from control, 6-OHDA, and chronic LD mice.

coefficient of variation (CV) of the instantaneous frequencies of sAP in ChIs from 6-OHDA mice, which was restored to sham control levels after chronic L-DOPA (sham: 0.36 ± 0.03, *n* = 81 neurons/26 mice; 6-OHDA: 0.57 ± 0.05, *n* = 80 neurons/29 mice; chronic-LD: 0.36 ± 0.04, *n* = 87 neurons/25 mice; p<0.0001, Kruskal-Wallis test; pairwise: sham vs. 6-OHDA, p=0.0004; sham vs. chronic-LD, p=0.8126, Dunn's multiple comparison test; *Figure 1D*). Similarly, we found that 6-OHDA lesion decreased the number of both bursts and pauses, while chronic L-DOPA treatment of lesioned mice increased the number of bursts and restored the number of pauses compared to sham control levels. The duration of each burst was not different between the three groups, whereas pause duration was longer in the 6-OHDA and shorter in the chronic-LD groups. When the number of bursts and pauses were normalized to the number of sAP in each group, there was no significant difference between groups, suggesting that these irregularities of firing pattern changed in parallel with altered firing rate (*Figure 1—figure supplement 2A–F*).

We next assessed evoked AP firing as a measure of ChI excitability with current clamp recordings during current injection steps from 0 to 300 pA. Similar to the changes in sAP frequency, ChI from 6-OHDA lesioned mice fired fewer evoked AP than ChI from sham animals, indicating that dopamine depletion decreased excitability. Chronic treatment with L-DOPA, however, only restored excitability back to sham levels (at 300 pA; sham: 35.8 ± 0.8 Hz, *n* = 11 neurons/4 mice; 6-OHDA: 26.1 ± 1.2 Hz, *n* = 16 neurons/5 mice; chronic-LD: 33.4 ± 1.6 Hz, *n* = 18 neurons/5 mice; p<0.0001 for interaction, Two-way ANOVA; pairwise group means: sham vs. 6-OHDA, p=0.0003; sham vs. chronic-LD, p=0.4173, Tukey's multiple comparison test; *Figure 1E*).

Our group previously reported that chronic treatment with L-DOPA in a genetic model of dopamine depletion, but not in dopamine-intact mice, rendered ChI hyper-responsive to bath-applied dopamine during recording (*Ding et al., 2011*). Consistent with this, we found that bath exposure to dopamine increased spontaneous ChI firing rate only in brain slices from the 6-OHDA and chronic-LD groups, indicating that the effect of dopamine is biased towards excitation by dopamine depletion (dopamine-mediated change in sAP frequency: sham: 0.6 ± 0.3 Hz, *n* = 7 neurons/3 mice, p=0.056; 6-OHDA: 2.4 ± 0.4 Hz, *n* = 7 neurons/2 mice, p=0.0012; chronic-LD: 1.7 ± 0.4 Hz, *n* = 10 neurons/3 mice, p=0.0026, Paired t test; *Figure 1F,G*).

Thus, DA depletion significantly decreases the spontaneous firing rate, regularity of spiking and excitability of ChI, while increasing the firing rate in response to dopamine. Chronic treatment of

dopamine-depleted animals with L-DOPA restores the key parameters of regularity and excitability, but increases firing frequency beyond that of control animals.

## Electrophysiological characteristics and synaptic input to striatal ChIs

To determine whether changes in basic electrophysiological characteristics of ChIs could explain observed alterations in spontaneous activity of the cells, we performed whole-cell recordings. For voltage-current dependence, resting membrane potential, input resistance, and membrane capacitance (*Figure 1—figure supplement 3A–D*) measurements, tetrodotoxin (TTX) was included during recording to prevent spontaneous APs. Likewise, ChI were briefly hyperpolarized before the current ramping protocol for rheobase determination (*Figure 1—figure supplement 3E,F*). There were no differences in these properties between the experimental groups. Similarly, analysis of the waveforms of spontaneous or evoked action potentials (*Figure 1—figure supplement 3G–L*) generally revealed no differences in peak amplitude, half width, rise time, decay time, threshold, and latency to first spike; however, the sAP threshold potential was more positive in the +chronic-LD group vs. 6-OHDA and the rise time of evoked APs was longer in the 6-OHDA group vs. sham.

Several reports have shown that changes in neuronal excitability correlate with remodeling of their dendritic arbors (*Al-Muhtasib et al., 2018*; *Cazorla et al., 2012*; *Fieblinger et al., 2018*). We assessed this with Sholl analysis of three dimensional neuritic arbor reconstructions from streptavidin labeled ChIs that were filled with biotin during recording (*Figure 2A*). ChIs from both the 6-OHDA and chronic-LD groups had more intersections with Sholl radii as a function of distance from the soma and more total intersections (total intersections with Sholl radii; sham: 217 ± 18, *n* = 13 neurons/7 mice; 6-OHDA: 372 ± 46, *n* = 28 neurons/14 mice; chronic-LD: 420 ± 72, *n* = 19 neurons/10 mice; p=0.0219, Kruskal-Wallis test; pairwise: sham vs. 6-OHDA, p=0.0311; sham vs. chronic-LD, p=0.0454, Dunn's multiple comparison test; *Figure 2B,D*). This was accompanied by greater total neurite length and a larger ending radius (total neurite length; sham: 3639 ± 327 μm; 6-OHDA: 6450 ± 844 μm; chronic-LD: 7515 ± 1421 μm; p=0.0166, Kruskal-Wallis test; pairwise: sham vs. 6-OHDA, p=0.0234; sham vs. chronic-LD, p=0.0364, Dunn's multiple comparison; *Figure 2C,E*), although the number of primary branches and ramification index were not different (*Figure 2F,G*). These data suggest that dopaminergic lesion induced a remodeling of ChI neurites that was not reversed with chronic L-DOPA treatment.

We next investigated whether changes in excitatory and inhibitory synaptic inputs followed observed adaptations in dendritic morphology. Spontaneous inhibitory postsynaptic currents (sIPSC) frequency was increased in the chronic-LD group compared to sham mice, while sIPSC amplitude was unchanged (sIPSC amplitude; sham: 26.9 ± 2.2 pA, *n* = 9 neurons/5 mice; 30.0 ± 3.8 pA, *n* = 12 neurons/6 mice; chronic-LD: 23.2 ± 2.1 pA, *n* = 7 neurons/3 mice; p=0.1824, Kruskal-Wallis test; sIPSC frequency; sham: 2.4 ± 0.6 Hz; 6-OHDA: 3.3 ± 0.7 Hz; chronic-LD: 5.5 ± 0.9 Hz; p=0.0240, Kruskal-Wallis test; pairwise: sham vs. 6-OHDA, p=0.8769; sham vs. chronic-LD: p=0.0202, Dunn's multiple comparison test; *Figure 2H,I*). In contrast, neither frequency nor amplitude of spontaneous excitatory postsynaptic currents (sEPSC) was changed by the treatments (sEPSC amplitude; sham: 24.2 ± 2.8 pA, *n* = 8 neurons/4 mice; 6-OHDA: 23.2 ± 1.3 pA, *n* = 16 neurons/7 mice; chronic-LD: 21.8 pA, *n* = 11 neurons/6 mice; p=0.7949, Kruskal-Wallis test; sEPSC frequency; sham: 0.2 ± 0.1 Hz; 6-OHDA: 0.4 ± 0.1 Hz; chronic-LD: 0.1 ± 0.03 Hz; p=0.2276, Kruskal-Wallis test; *Figure 2J,K*). However, due to the rarity of spontaneous excitatory events, we are likely underpowered for a definite conclusion about the lack of change in sEPSCs (*Bennett and Wilson, 1999*). These results demonstrate that L-DOPA treatment of DA lesioned mice increased synaptic connectivity of ChIs with GABA neurons, despite dendritic remodeling in both DA lesioned and chronic-LD groups.

## Restoration of DA depletion-induced decrease of HCN current by chronic L-DOPA

Examination of the whole-cell recording traces from ChIs of the three groups revealed marked differences in the kinetics of voltage change following the action potential. In the 6-OHDA group, the cell returned to the threshold firing potential at a significantly slower rate, while in the chronic-LD group, the hyperpolarization that followed the AP appeared to be lower in amplitude (*Figure 3A*). This prompted us to next look at the activity of the channels known to mediate the afterhyperpolarization (AHP) currents that regulate the rate of spontaneous firing of striatal ChIs.

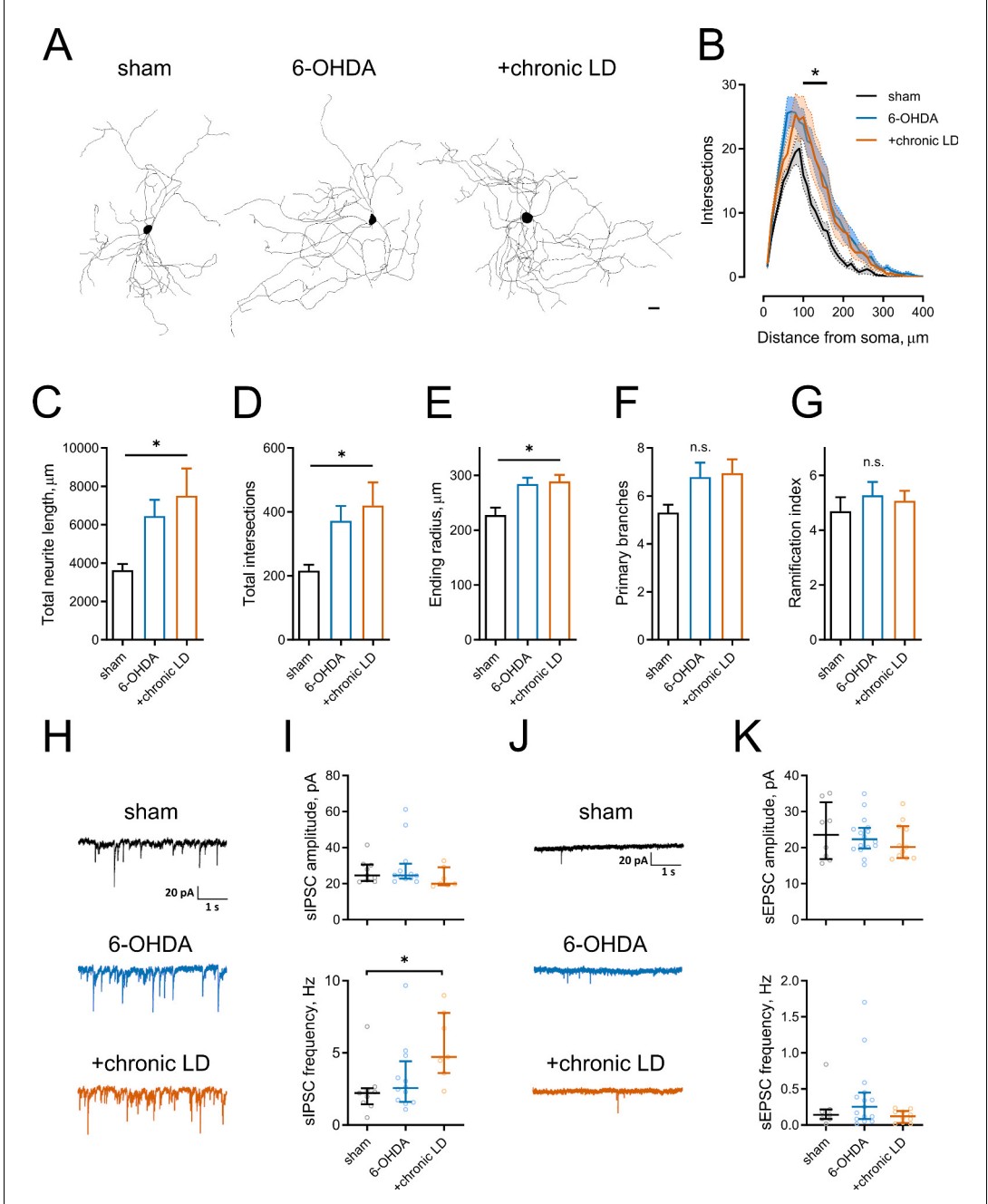

**Figure 2.** Morphological parameters and synaptic inputs in ChIs from control, 6-OHDA lesioned and chronic LD mice. (A–G) Sholl analysis of ChI morphology. Cells were filled with biocytin during physiological recordings, fixed, and imaged for biocytin labeling by confocal microscopy. (A) Representative maximum projection images from three dimensional reconstructions of ChIs, scale bar = 20 μm. (B) Sholl analysis of reconstructed ChIs, solid line denotes mean intersections at indicated distance from soma, shaded area shows SEM, p<0.05 (*) for sham vs. 6-OHDA and sham vs. chronic LD, Dunnett's multiple comparison test following two-way ANOVA, p=0.0074 for interaction between treatment group and distance from soma. (C–G) DA lesion caused significant increase in total dendrite length (C), total number of intersections (D) and ending radius (E), whereas the number of primary dendrites (F) and ramification index (G) were similar between the groups, p<0.05 (*) 6-OHDA and +chronic-LD vs. Sham, Kruskal-Wallis test with Dunn's multiple comparison. For A-G sham n = 13 neurons/7 mice, 6-OHDA n = 28 neurons/14 mice, chronic-LD n = 19/10 mice. (H and J) Representative traces of spontaneous inhibitory postsynaptic currents (sIPSCs) and excitatory postsynaptic currents (sEPSCs) of ChIs from sham, 6-OHDA and chronic LD mice. (I) Amplitudes of sIPSCs were similar, but their frequency was increased in chronic L-DOPA group. Sham n = 9 neurons/5 mice, 6-OHDA n = 12 neurons/6 mice, chronic-LD n = 7 neurons/3 mice. p<0.05 (*) by Kruskal-Wallis test followed by Dunn's multiple comparison test. (K) There were no changes in amplitude or frequency of sEPSCs. Sham n = 8 neurons/4 mice, 6-OHDA n = 16 neurons/7 mice, chronic-LD n = 11 neurons/6 mice.

*Figure 2 continued on next page*

*Figure 2 continued*

The online version of this article includes the following source data for figure 2:

**Source data 1.** Individual neuron data and statistics for all panels.

Hyperpolarization-activated cyclic nucleotide-gated (HCN) channels are essential for the cell-autonomous spontaneous firing of ChIs (*Ferreira et al., 2014*; *Oswald et al., 2009*; *Wilson, 2005*). Accordingly, bath application of an HCN channel antagonist ZD7288 (25 µM) significantly decreased ChI sAP frequency in our slice preparation (*Figure 3B*). To characterize HCN channel activity, we measured the characteristic voltage sag induced by hyperpolarizing current injection in the current clamp mode (*Figure 3C*) and also directly isolated ZD7288 sensitive HCN currents ($I_h$) in the voltage clamp mode (*Figure 3E,F*). ChIs from 6-OHDA mice exhibited smaller voltage sag amplitudes and decreased $I_h$ than either sham or chronic-LD mice, whereas there was no difference in these parameters in ChIs from chronic-LD mice compared to sham (sag amplitude at −250 pA; sham: −16.6 ± 0.8 mV, $n$ = 39 neurons/18 mice; 6-OHDA: −13.4 ± 0.9 mV, $n$ = 23 neurons/11 mice; chronic-LD: −17.9 ± 1.3 mV, $n$ = 30 neurons/12 mice, p=0.0255 for group effect, two-way ANOVA; pairwise comparison at −250 pA: sham vs. 6-OHDA, p=0.0266; sham vs. chronic-LD, p=0.6781, Tukey's multiple comparison test; *Figure 3D*; $I_h$ density at −100 mV; sham: −0.94 ± 0.09 pA/pf, $n$ = 11 neurons/6 mice; 6-OHDA: −0.59 ± 0.10 pA/pF, $n$ = 12 neurons/6 mice; chronic-LD: −1.01 ± 0.15 pA/pF, $n$ = 13 neurons/7 mice; p=0.0025 for interaction, two-way ANOVA; pairwise comparison at −100 mV: sham vs. 6-OHDA, p=0.0464; sham vs. chronic-LD, p=0.9078, Tukey's multiple comparison test; *Figure 3F*). Furthermore, Boltzmann fits of normalized voltage dependences of HCN currents showed a more negative half activation voltage in ChIs from 6-OHDA mice compared to sham and chronic-LD animals ($V_{50}$; sham: −88.8 ± 0.9 mV; 6-OHDA: −95.6 ± 2.5 mV; chronic-LD: −91.0 ± 1.4 mV, one-way ANOVA, p=0.0312; pairwise: sham vs. 6-OHDA, p=0.0287; sham vs. chronic-LD, p=0.6601; *Figure 3G*), suggesting that the gating properties or the expression profile of HCN isoforms were altered by 6-OHDA lesion and restored by chronic L-DOPA exposure (*Simeone et al., 2005*; *Wang et al., 2001*; *Zolles et al., 2006*).

We therefore next assessed whether the ChI-specific gene expression profiles of HCN channel isoforms were changed by DA depletion or chronic L-DOPA treatment using bitransgenic *Chat-Cre: Rpl22HA* (ribotag) mice which express a tagged ribosomal subunit only in cholinergic neurons (*Sanz et al., 2009*). We subjected these mice to the same DA depletion and chronic L-DOPA treatment paradigm and then collected tagged ribosome-bound mRNA for RT-qPCR analysis. mRNA was harvested 20 hr after the last injection of saline or L-DOPA, reflecting steady-state gene expression levels. This technique yielded ~38 fold enrichment of cholinergic neuron specific genes compared to total striatal mRNA (*Chat:Actb*; ribotag: 0.42 ± 0.01 vs. input: 0.01 ± 0.001). Consistent with previous findings, *Hcn2* was the most abundantly expressed HCN isoform in ChIs (*Figure 3H*) while *Hcn3* and *Hcn4* mRNA were enriched in ChIs compared to total striatal input (not shown), though there was no statistically significant difference in mRNA levels of *Hcn1-4* between any groups. The trafficking and gating of HCN channels is regulated by tetratricopeptide repeat-containing Rab8b-interacting protein (TRIP8b; gene name *Pex5l*), an auxiliary β subunit expressed in neurons (*Bankston et al., 2012*; *Lewis et al., 2011*). TRIP8b mRNA levels were significantly reduced in ChIs from 6-OHDA mice (*Pex5l:Actb* ratio; sham: 0.066 ± 0.003, $n$ = 6 samples/20 mice; 6-OHDA: 0.053 ± 0.003, $n$ = 4 samples/17 mice; chronic-LD: 0.060 ± 0.003, $n$ = 6 samples/22 mice; p=0.0462, Kurskal-Wallis test; pairwise: sham vs. 6-OHDA, p=0.0474; sham vs. chronic-LD, p=0.6087, Dunn's multiple comparisons test; *Figure 3H*), which may contribute to the decrease in their HCN current. Together, these findings indicate that the decrease in firing rate of ChIs from 6-OHDA mice could be due to decreased HCN function. However, as HCN activity and expression of its subunits was restored to sham levels in the chronic-LD group, changes in $I_h$ alone cannot account for the accompanying overshoot in ChI firing caused by chronic L-DOPA treatment.

## Persistent decrease in medium afterhyperpolarization currents

The AHP following each ChI action potential is mediated by potassium channels that are activated during cell depolarization and concomitant $Ca^{2+}$ influx (*Bennett et al., 2000*; *Goldberg and Wilson, 2005*; *Tubert et al., 2016*). At low frequency spiking of ChIs, such as during their cell-autonomous

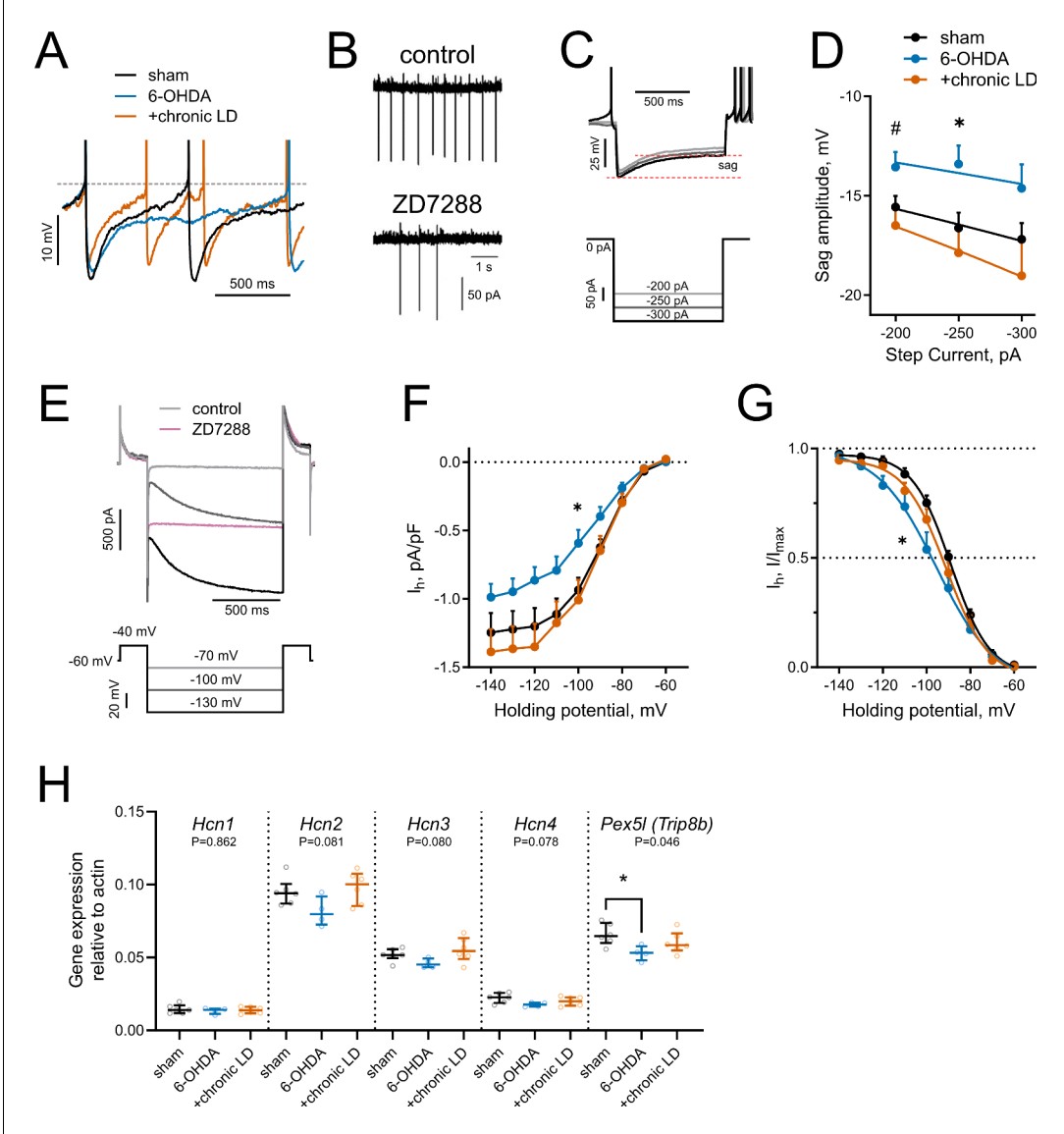

**Figure 3.** HCN-mediated currents are decreased by 6-OHDA lesion. (**A**) Representative perforated-patch recordings of sAP from sham, 6-OHDA and +chronic-LD groups. Dotted line represents a threshold potential, which was similar between the groups. Note the markedly slower rate of cell depolarization after the action potential in the ChIs from 6-OHDA group and decreased amplitude of the AHP in a neuron from the +chronic-LD group. (**B**) Representative cell-attached recordings of ChI activity before and after treatment with HCN channel blocker ZD7288 (25 µM). (**C**) Current-clamp protocol (lower) and representative recordings (upper) showing voltage sag, a characteristic of HCN channel activation. (**D**) Quantification of sag amplitude at different current steps. Sham n = 39 neurons/18 mice, 6-OHDA n = 23 neurons/11 mice, chronic-LD n = 30 neurons/12 mice; p<0.05 (*) for 6-OHDA vs. two other groups, p<0.05 (#) 6-OHDA vs. chronic-LD at indicated current by Tukey's multiple comparisons test following repeated measures two-way ANOVA. (**E**) Voltage-clamp protocol (lower) and representative ZD7288-sensitive ($I_h$) current (upper). (**F**) $I_h$ density was decreased in ChIs from DA-depleted mice. Sham n = 11 neurons/6 mice, 6-OHDA n = 12 neurons/6 mice, chronic-LD n = 13 neurons/7 mice; p<0.05 (*) for sham vs. 6-OHDA at −100 mV by Tukey's multiple comparison following repeated measures two-way ANOVA. (**G**) Boltzmann fits of normalized $I_h$ densities. (Same N as in F); p<0.05 (*) for V50 values of sham vs. 6-OHDA by Tukey's multiple comparison following one-way ANOVA. (**H**) ChI-specific gene expression of *Hcn1-4* isoforms and *Pex5l (Trip8b)* measured by RT-qPCR from striatal mRNA immunoprecipitated from *Chat-Cre:Rpl22*[HA] (ribotag) mice treated as indicated. Target mRNA levels were normalized to β-actin. Sham n = 6 samples/20 mice, 6-OHDA n = 4 samples/17 mice, chronic-LD n = 6 samples/22 mice; P-values on graphs are for Kruskal-Wallis test, p<0.05 (*) with Dunn's multiple comparisons test.

*Figure 3 continued on next page*

*Figure 3 continued*

The online version of this article includes the following source data for figure 3:

**Source data 1.** Individual neuron data and statistics for all panels.

activity, small conductance calcium-activated potassium channels (SK) mediate a medium-duration afterhyperpolarization (mAHP). Conversely, prolonged high frequency firing of ChIs activates a slow afterhyperpolarization (sAHP) that can last several seconds (*Goldberg et al., 2009*; *Wilson and Goldberg, 2006*).

To assess changes in AHP currents, first we applied a 300 ms depolarizing pulse from −60 mV to 10 mV, which induced a train of action potentials that was followed by a characteristic tail current lasting seconds after the end of the voltage pulse (*Figure 4A*). We measured the current during the *peak* response, which occurred within the first 200 ms following the offset of the voltage pulse, and during the long tail at 1000 ms (*late* response) (*Bennett et al., 2000*; *Sanchez et al., 2011*; *Wilson and Goldberg, 2006*). Both *peak* and *late* currents were inhibited by the non-selective $K^+$ channel blocker barium (200 μM) (*Figure 4A*), though their amplitudes were unchanged in ChIs from 6-OHDA and chronic-LD mice compared to the sham group (peak $Ba^{2+}$-sensitive current density; sham: 0.31 ± 0.06 pA/pF, *n* = 9 neurons/3 mice; 6-OHDA: 0.42 ± 0.03 pA/pF, *n* = 10 neurons/3 mice; chronic-LD: 0.51 ± 0.14 pA/pF, *n* = 9 neurons/3 mice; p=0.0873, Kruskal-Wallis test; late $Ba^{2+}$-sensitive current density; sham: 0.09 ± 0.02 pA/pF; 6-OHDA: 0.15 ± 0.02 pA/pF; chronic-LD: 0.15 ± 0.02 pA/pF; p=0.0756, Kruskal-Wallis test; *Figure 4B*). Next, to assess the currents that mediate mAHP more selectively, we used a shorter depolarization protocol, stepping from −60 mV to 0 mV for 100 ms (*Figure 4C*). As SK channels are the main mediators of mAHP currents, we used apamin (100 nM) to selectively block this channel (*Goldberg and Wilson, 2005*; *Wilson and Goldberg, 2006*). Apamin decreased *peak* AHP current without affecting the *late* component and revealed that SK channel activity was significantly decreased in ChIs from both 6-OHDA and chronic-LD groups compared to sham (peak apamin-sensitive current density; sham: 0.25 ± 0.03 pA/pF, *n* = 17 neurons/5 mice; 6-OHDA: 0.12 ± 0.03 pA/pF, *n* = 14 neurons/4 mice; chronic-LD: 0.13 ± 0.04 pA/pF, *n* = 14 neurons/4 mice; p=0.0094, Kruskal-Wallis test; pairwise: sham vs. 6-OHDA, p=0.0206; sham vs. chronic-LD, p=0.0381, Dunn's multiple comparison test; *Figure 4D*).

To determine whether changes in mAHP current were due to decreased expression of SK channels, we measured mRNA levels of SK1-3 (gene name *Kcnn1-3*) in ChIs from *Chat-Cre:Rpl22^HA* mice. There was no difference in the mRNA expression of these SK isoforms, suggesting that changes in mAHP current were not mediated by transcriptional regulation of these channels (*Figure 4E*).

## Partial inhibition of HCN and SK channels recapitulates changes in ChI firing rate caused by DA depletion and chronic L-DOPA treatment

Based on the changes in $I_h$ and mAHP currents presented above, we hypothesized that altered activity of HCN and SK channels might be responsible for decreased spontaneous activity in 6-OHDA and increased spontaneous activity in chronic-LD mice (*Figure 5A*). To model these changes pharmacologically, we first established the concentrations of ZD7288 and apamin that provide ~50% inhibition of HCN- and SK-mediated currents observed in ChIs from 6-OHDA and chronic-LD mice (ZD7288: EC50 = 1.4 μM, $R^2$ = 0.87, *n* = 8 neurons/2 mice; apamin: EC50 = 0.9 nM, $R^2$ = 0.88, *n* = 6 neurons/2 mice; *Figure 5B,C*). Next, using slice preparations from control mice (no sham surgery), we performed cell-attached recordings of ChI activity before and after bath application of 1 nM apamin either alone (to mimic decreased SK but normal HCN activity seen in chronic-LD group) or in combination with 1 μM ZD7288 (to mimic inhibition of both SK and HCN channels in 6-OHDA group). Partial blockade of both channels decreased sAP frequency (baseline: 1.9 ± 0.3 Hz; apamin + ZD7288: 0.9 ± 0.2 Hz, *n* = 8 neurons/3 mice; p=0.0078, Wilcoxon signed rank test; *Figure 5D,E*), increased the coefficient of variation of sAP frequency (baseline: 0.28 ± 0.04; apamin + ZD7288: 0.74 ± 0.15; p=0.0156, Wilcoxon signed rank test; *Figure 5F,G*) and phenocopied changes in bursting and pausing of ChIs in the 6-OHDA group (*Figure 1—figure supplement 2G–L*). Similarly, the features of ChI firing in the chronic-LD group were recapitulated with partial SK current blockade, including increased sAP frequency (baseline: 1.8 ± 0.2 Hz; apamin: 2.7 ± 0.3 Hz, *n* = 10 neurons/3 mice; p=0.0020, Wilcoxon signed rank test; *Figure 5D,E*), no change in coefficient of variation of

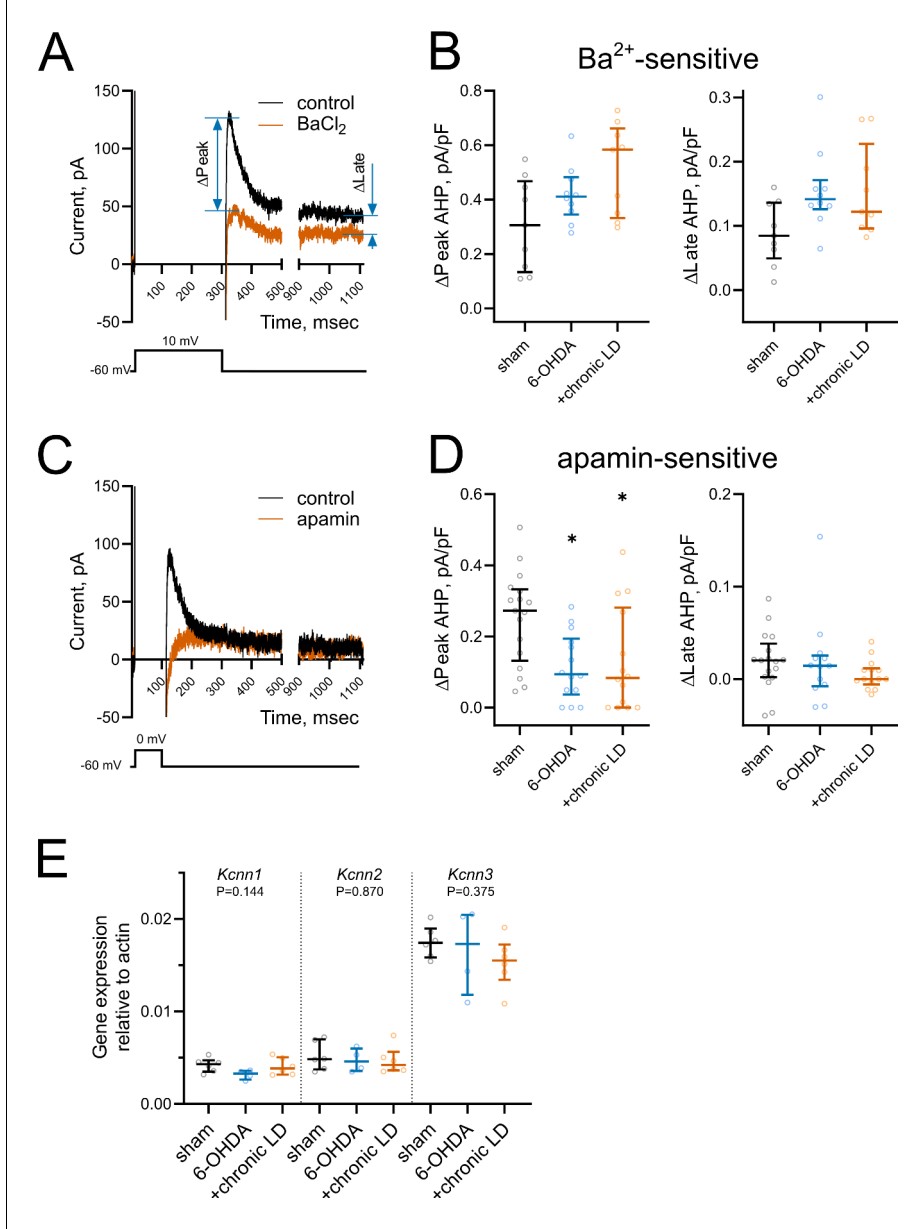

**Figure 4.** Changes in medium and slow afterhyperpolarization (AHP) currents. (**A and C**) Representative voltage-clamp recording of (**A**) $Ba^{2+}$-sensitive and (**C**) apamin-sensitive AHP currents before and after drug application (upper), and corresponding voltage protocols (lower). (**B**) $BaCl_2$ (200 µM) blocked both *peak* and *late* phases of the current but the magnitude of the changes was similar in all groups. Sham n = 9 neurons/3 mice, 6-OHDA n = 10 neurons/3 mice, chronic-LD n = 9 neurons/3 mice. (**D**) Using a depolarization protocol to recruit primarily mAHP currents, apamin (100 nM) decreased the *peak* current amplitude without altering the *late* stage of the AHP current. Sham n = 17 neurons/5 mice, 6-OHDA n = 14 neurons/4 mice, chronic-LD n = 14 neurons/4 mice; p<0.05 (*), Kruskal-Wallis test with Dunn's multiple comparison. (**E**) ChI-specific gene expression of *Kcnn1-3* (SK1-3) isoforms measured by RT-qPCR as in *Figure 3H*. Target mRNA levels were normalized to β-actin. There were no significant differences between the groups by Kruskal-Wallis test; sham n = 6 samples/20 mice, 6-OHDA n = 4 samples/17 mice, chronic-LD n = 6 samples/22 mice.

The online version of this article includes the following source data for figure 4:

**Source data 1.** Individual neuron data and statistics for all panels.

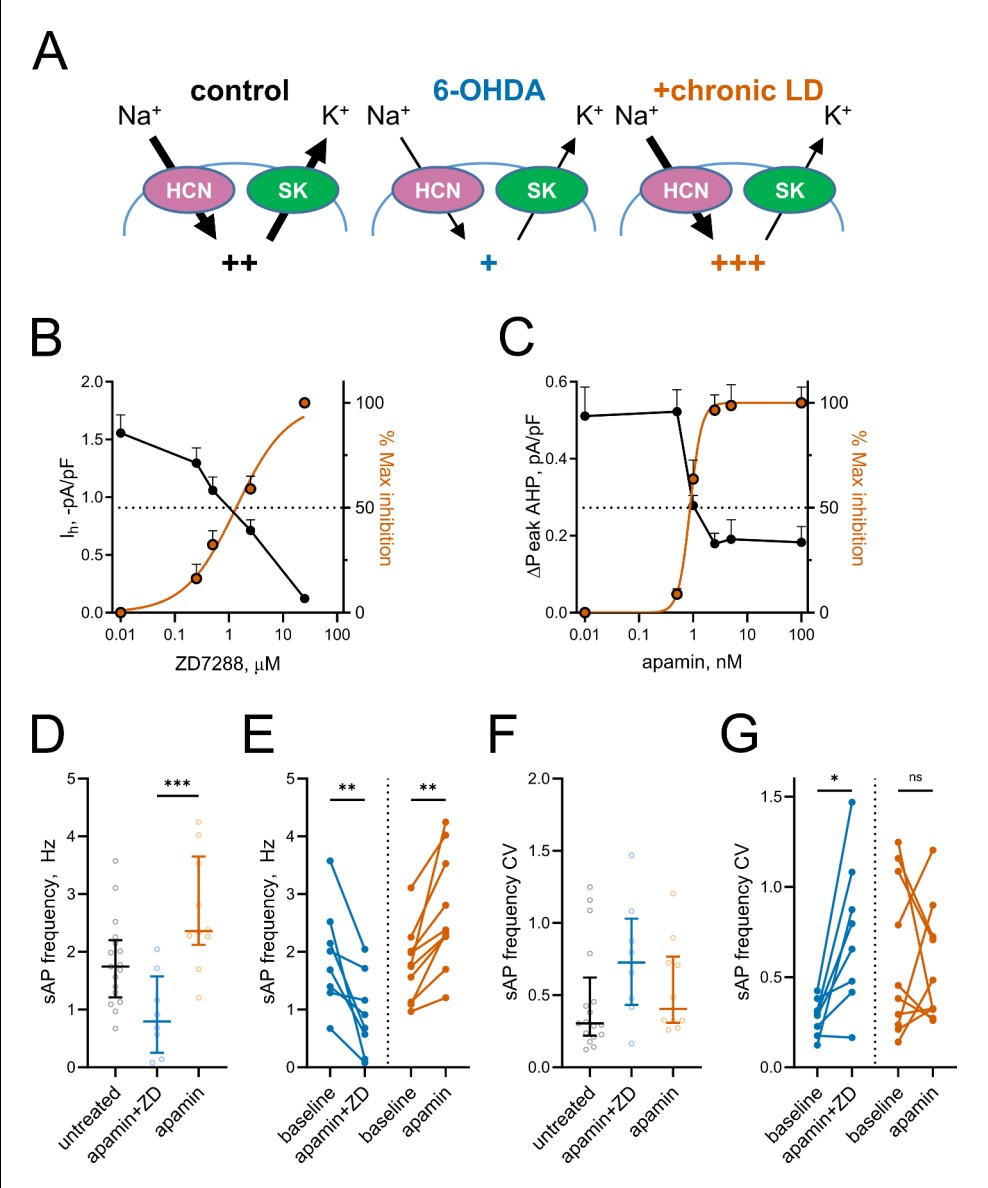

**Figure 5.** Partial inhibition of HCN and SK channels is sufficient to mimic changes in ChI activity after lesion and chronic L-DOPA treatment. (A) Proposed changes in HCN and SK currents in ChIs from different treatment groups and their effect on spontaneous firing rates (+). (B) Dependence of $I_h$ density (as measured on **Figure 3E**) on ZD7288 concentration. The orange curve represents fit of the data with the equation $Y = 100(X^{Slope})/(IC_{50}^{Slope} + X^{Slope})$. Hill slope and $IC_{50}$ are 0.9 and 1.4 µM, correspondingly; n = 8 neurons/2 mice. (C) Dependence of mAHP current density (as measured on **Figure 4C**) on apamin concentration. Hill slope and $IC_{50}$ are 4 and 0.9 nM, correspondingly; n = 6 neurons/2 mice.(D–G) Partial SK and HCN channel blockade reproduced changes in the sAP rate and coefficient of variation of 6-OHDA and +chronic-LD groups. Although on average the decrease and increase in sAP frequency caused by apamin+ZD and apamin alone, respectively, did not reach statistical significance (D), apamin+ZD reliably decreased and apamin alone increased baseline sAP firing rate in individual ChIs (E). Likewise, the differences in the median coefficient of variation did not reach statistical significance (F), however, apamin+ZD increased CV over baseline for most ChIs, whereas apamin alone did not change the baseline CV (G). Untreated n = 17 neurons/6 mice, apamin+ZD n = 8 neurons/3 mice, apamin n = 10 neurons/3 mice; In panel D, p<0.001 (***) by Kruskal-Wallis test with Dunn's multiple comparisons. In panels E and G, p<0.05 (*) and p<0.01 (**), Wilcoxon matched-pairs signed rank test.

The online version of this article includes the following source data for figure 5:

**Source data 1.** Individual neuron data and statistics for all panels.

sAP frequency (baseline: 0.60 ± 0.14; apamin: 0.55 ± 0.10; p=0.8457, Wilcoxon signed rank test; *Figure 5F,G*) and similar changes in the bursting/pausing patterns (*Figure 1—figure supplement 2*). Overall, our findings suggest that alterations in ChI spontaneous activity caused by DA depletion and subsequent chronic L-DOPA treatment can be reproduced by decreasing HCN- and SK- mediated currents.

## Discussion

Dysregulation of cholinergic neurotransmission is an important contributor to both the expression of PD symptoms and the adverse effects of DA replacement therapy. Here, we examined the consequences of DA loss and those of chronic treatment with L-DOPA on mouse ChI physiology. In addition to altered morphology, synaptic connectivity and responsiveness to DA receptor stimulation, we found significant alterations of cell-intrinsic properties of ChIs. In the DA depleted striatum, both HCN- and SK-mediated currents were diminished. HCN current reduction was accompanied by reduced TRIP8b, which regulates trafficking and gating of HCN channels, rather than changes in *Hcn* mRNA expression. Interestingly, chronic treatment of lesioned mice with L-DOPA restored HCN activity to sham levels while SK currents remained depressed. The pharmacological blockade of HCN and SK channels to mimic the DA depleted and chronic L-DOPA treated states recapitulated the changes in rate and pattern of ChI firing, providing new insights into molecular adaptations that follow striatal DA depletion in PD patients receiving L-DOPA therapy.

### Changes in ChI tonic activity following DA depletion and chronic L-DOPA treatment

Anticholinergic drugs have been effective in the treatment of PD symptoms, highlighting the importance of cholinergic neurotransmission to basal ganglia function. Early observations showed that the severity of PD symptoms in patients is worsened by drugs that increase cholinergic activity (*Duvoisin, 1967*), while more recent reports demonstrated that direct optogenetic and chemogenetic inhibition of ChIs improved DA depletion-mediated motor dysfunction (*Maurice et al., 2015*; *Tanimura et al., 2019*; *Ztaou et al., 2016*). Though the contribution of aberrant ChI neurotransmission to PD pathophysiology is unquestioned, conflicting results on the change in ChI activity caused by DA depletion have been reported. We show here that both spontaneous activity and excitability of dorsolateral striatal ChIs in 6-OHDA injected mice are significantly decreased. Similar changes were recently demonstrated using a genetic model with diphtheria toxin to induce dysfunction in dopaminergic neurons (*McKinley et al., 2019*). In contrast, other studies using similar protocols in rodents have shown no significant change in the tonic firing rate (*Ding et al., 2006*), and increased excitability of ChIs (*Maurice et al., 2015*; *Sanchez et al., 2011*; *Tubert et al., 2016*). Some of these discrepancies can be attributed to differences in animal species (mice vs. rats) or experimental conditions (DA-lesion protocol, composition of the recording solutions, etc.) but careful evaluation of the contribution of these differences is currently lacking.

Levels of acetylcholine in striatal tissue or CSF from PD patient samples are not different from controls (*Duvoisin and Dettbarn, 1967*; *Welch et al., 1976*), but these approaches cannot resolve differences in acetylcholine release due to altered synaptic activity. The lack of change in overall acetylcholine levels raises the question of whether the 'hypercholinergic state' in PD reflects an increase in basal cholinergic release, a shift in the balance between DA and acetylcholine levels (*McKinley et al., 2019*), or altered firing of ChIs in response to physiological stimuli. For example, striatal tonically-active neurons (putative ChIs) in non-human primates exhibit a pause response that is acquired after training in a classical conditioning task (*Aosaki et al., 1994*). In trained animals, the conditioned stimulus elicits a brief pause (~200 ms) in ChI tonic firing, followed by a brief rebound increase in firing rate. Lesion of the striatonigral system with 1-methyl-4-phenyl-1,2,3,6-tetrahydropyridine (MPTP) did not change the tonic firing rate of ChIs, but abolished this pause response in vivo (*Aosaki et al., 1994*). As this pause is thought to provide a 'window' to encode cortico-striatal plasticity (*Deffains and Bergman, 2015*), loss of this pause response and increased variability of ChI firing (*Figure 1D*; *McKinley et al., 2019*) may cause dysregulation of cortico-striatal neurotransmission and aberrant striatal plasticity. Additionally, the loss of DA may change the response of striatal neurons to acetylcholine as the degree of muscarinic and nicotinic acetylcholine receptor binding is altered in the brain tissue of PD patients (*Aubert et al., 1992*; *Joyce, 1993*; *Pimlott et al., 2004*)

and after experimental DA depletion (*Cremer et al., 2015*; *Joyce, 1991*). Combined with the dendritic remodeling observed in our results (*Figure 2*) and other studies (*Lozovaya et al., 2018*), and changes in the connectivity from ChIs to spiny projection neurons (*Salin et al., 2009*) that may drive changes in the regulation of neurotransmitter release from striatonigral synapses (*Borgkvist et al., 2015*), it is likely that the regulation of striatal output by ChIs is perturbed as a result.

Chronic L-DOPA treatment of 6-OHDA-lesioned mice increased the tonic firing of ChIs beyond the sham lesioned group (*Figure 1C*), while restoring many firing pattern abnormalities (*Figure 1—figure supplement 2*) and neuronal excitability (*Figure 1E*) back to sham levels. This may represent the cellular mechanism underlying the hypercholinergic state that contributes to motor dysfunction in PD patients who have undergone DA replacement therapy. Consistent with this idea, ablation of ChIs reduces LID (*Won et al., 2014*) and direct activation of ChIs increases LID (*Aldrin-Kirk et al., 2018*; *Bordia et al., 2016a*) in rodent PD models, although the effect of ChI modulation is complex (*Bordia et al., 2016a*; *Divito et al., 2015*; *Maurice et al., 2015*). It is important to note that these effects of long-term DA replacement require the parkinsonian state as chronic treatment with L-DOPA of DA-intact mice does not alter the tonic firing rate of ChI (*Ding et al., 2011*), suggesting that the exaggerated response to DA receptor activation following DA depletion underlies the adaptations that increase in tonic firing of ChI (*Figure 1G*). In aggregate, these studies suggest that there is an interaction between an increased rate and improper pattern/context of ChI firing in PD and after L-DOPA treatment. Reinstating both of these aspects of cholinergic neurotransmission may be critical for restoring the functional behavior.

In slice preparations, ChIs fire spontaneously and generate a variety of spiking patterns and firing frequencies autonomously. Though excitatory and inhibitory inputs to ChIs are present, these are largely silent as the firing rate and pattern of ChIs are unaffected by blockade of AMPA, NMDA, GABA$_A$, D1, D2, or muscarinic receptors (*Bennett and Wilson, 1999*). Consistent with this, the changes in tonic firing rate caused by DA depletion and chronic L-DOPA treatment persisted in the presence of blockers of ionotropic glutamate and GABA receptors (*Figure 1C*), indicating that these changes are intrinsic to ChIs. Furthermore, the increases in dendritic branching and GABAergic inputs onto ChIs after chronic L-DOPA treatment (*Figure 2*), which likely represent compensatory remodeling of striatal circuitry, did not have significant effect on the tonic firing of ChIs in our preparation. Together with altered responsiveness to DA receptor stimulation (*Figure 1G*), these synaptic adaptations may, however, play important roles in vivo, which will be addressed in future studies.

## DA depletion and chronic L-DOPA treatment change HCN activity

HCN channels are essential for the tonic firing of ChIs and serve to depolarize neurons back towards spike threshold during the hyperpolarization that follows an action potential. Inhibition of $I_h$ decreases the rate and increases irregularity of tonic firing in ChIs (*Bennett et al., 2000*; *Deng et al., 2007*; *McKinley et al., 2019*; *Zhao et al., 2016*). In line with this, we found that DA depletion reduced $I_h$ and shifted the gating for channel activation to more negative potentials (*Figure 3*), which was accompanied by slower and irregular firing. By contrast, chronic treatment of DA depleted mice with L-DOPA restored $I_h$ to the level of sham-lesioned animals, though the firing rate exceeded that of ChI from sham animals. These findings are consistent with a recent study using a targeted diphtheria toxin model of DA depletion which found a similar decrease of $I_h$ that was mediated by the transcriptional downregulation of specific *Hcn* isofroms (*McKinley et al., 2019*). Although we did not observe statistically significant changes in *Hcn* mRNA levels, we found that DA depletion significantly decreased Trip8b mRNA levels in ChIs (*Figure 3H*). Trip8b is expressed in neurons and regulates surface expression and trafficking of HCN channels. Knockdown of Trip8b in vivo causes the mis-localization of HCN channels and reduces $I_h$ in hippocampal CA1 neurons (*Piskorowski et al., 2011*). Our results confirm a previous report of reduction in Trip8b mRNA after DA depletion in the external globus pallidus (GPe) in conjunction with decreased $I_h$ (*Chan et al., 2011*). In our study, chronic treatment with L-DOPA reversed the decrease in Trip8b suggesting that striatal DA, including the levels achieved with once daily administration of L-DOPA, may maintain HCN activity by regulating Trip8b expression. This effect lasts longer than pharmacokinetic availability of L-DOPA (*Abercrombie et al., 1990*) since the recordings were performed at least 20 hr after the last L-DOPA dose, reflecting the long-term adaptation to chronic L-DOPA exposure rather than the immediate pharmacologic actions of L-DOPA itself. A possible mechanism linking DA depletion with changes in HCN biophysical properties may involve the regulation of cAMP production by DA

receptor signaling (*Greengard, 2001*). Binding of cAMP to HCN channels, which can also be regulated by Trip8b (*Hu et al., 2013*; *Saponaro et al., 2014*), shifts their activation kinetics towards more positive potentials and increases channel opening kinetics (*Wainger et al., 2001*). We found that HCN activation was shifted to more hyperpolarized potentials in ChIs from DA depleted mice and restored by chronic L-DOPA treatment (*Figure 3G*), although whether these changes are mediated by decreased cAMP availability or other DA-dependent mechanisms remains to be tested.

The somatodendritic localization of HCN channels determines their effect on neuronal activity. In ChIs, HCN channels located at the soma depolarize neurons towards threshold potential and increase the firing rate of tonically-active neurons (*Bennett et al., 2000*). However, high expression of HCN channels in the distal dendrites of cortical or hippocampal pyramidal neurons dampens temporal summation of excitatory synaptic input to enforce dendritic integration, decreasing excitability. Accordingly, deletion of HCN1, HCN2, Trip8b, or pharmacological inhibition with ZD7288 increases dendritic summation in these neurons and increases dendritic and cellular excitability (*Harnett et al., 2015*; *Lewis et al., 2011*). In layer five cortical pyramidal neurons, this property is acquired during postnatal dendritic maturation where the attenuation of dendritic excitability follows the trafficking and concentration of HCN channels in the distal apical dendrites (*Atkinson and Williams, 2009*). The 'maturity' of the newly formed ChI dendritic arbors in dopamine-depleted and L-DOPA treated mice (*Figure 2A*) in terms of HCN channel localization and function may dictate whether they enhance or inhibit excitatory synaptic input onto ChIs in vivo. The downregulation of Trip8b mRNA in ChIs following dopamine depletion (*Figure 3H*) suggests that HCN channels may not be efficiently trafficked to dendritic sites which would increase dendritic excitability despite decreased tonic activity.

## DA depletion decreases SK current

During tonic firing of ChIs, calcium entry following each action potential induces outward potassium currents mediated by apamin-sensitive SK-channels (*Goldberg and Wilson, 2005*) and the kinetics of this afterhyperpolarization current determines the firing rate of ChIs (*Bennett et al., 2000*). We found decreased SK current in ChIs from DA depleted mice both with or without L-DOPA treatment, consistent with smaller AHP amplitudes. Although the reduction in SK current was the same for both groups, only parkinsonian mice treated chronically with L-DOPA showed higher firing rates, indicating that in the 6-OHDA group the decrease in firing rate from HCN current loss cannot be overcome by the increase in firing rate expected to result from reduction in SK current. The mRNA levels of SK channel isoforms were not different between groups indicating that the changes in channel activity are not directly mediated by their gene expression. In ChI, SK channel activation is coupled to calcium entry through $Ca_v2.2$ (N-type) channels, which are strongly inhibited by dopamine D2 and muscarinic M2/M4 receptors (*Goldberg and Wilson, 2005*; *Yan et al., 1997*; *Yan and Surmeier, 1996*). The reduction of SK current we observed could indicate persistent dysregulation of $Ca_v2.2$ channels or the G-protein signaling pathways to which they are coupled. Interestingly, modulation of $Ca_v2.2$-mediated $Ca^{2+}$ currents by M4 autoreceptors, but not D2 receptors, is diminished following dopamine depletion through the downregulation of G-protein signaling (*Ding et al., 2006*). This was proposed to prevent a decrease in SK current by M4 activation, though no direct measurements of SK current were made. Whether decreased SK channel activity is caused by perturbed auto-receptor signaling remains to be tested, although muscarinic antagonists did not influence spontaneous firing of ChIs in a previous study (*Bennett and Wilson, 1999*).

As with HCN channels, the subcellular location and channel subtype dictates the contribution of SK channels to neuronal activity. Midbrain dopaminergic neurons express SK1-3 subunits, with somatically localized SK3 channels being the dominant subtype and SK2 channels localized to distal dendrites. As with the ChI in our study, blockade of SK channels with apamin increases the rate of tonic activity, though the concentration of apamin used was higher (200 nM vs. 1 nM). The ability of apamin to increase firing rate is occluded in neurons from SK3[-/-], but not SK2[-/-] mice. However, this high concentration of apamin still caused firing irregularities in SK3[-/-] that were attributed to blocking dendritic SK2 (*Deignan et al., 2012*). As SK3 channels are the most abundantly expressed subtype in ChI, they likely play similar role in regulating tonic firing rate (*Figure 4E*), though the somatodendritic distribution SK channel subtypes in ChI is unknown.

## Pharmacological blockade of HCN and SK channels in ChIs mimics firing patterns caused by DA depletion and chronic L-DOPA treatment

Several changes in spontaneous firing of ChIs in our animal models were recapitulated by pharmacological inhibition of HCN and SK channels in striatal slices from control animals, including altered spontaneous activity and rhythmicity. Critically, we were able to titrate the concentrations of antagonists to resemble the degree of channel inhibition caused by DA depletion and chronic L-DOPA treatment. Consistent with previous studies, a saturating concentration of apamin caused ChIs to transition into burst firing mode, characterized by increased firing frequency within bursts, long pauses between bursts and decreased average firing frequency (data not shown and *Bennett et al., 2000*; *Yorgason et al., 2017*). However, when SK channels were only partially inhibited using an $IC_{50}$ concentration of apamin, the average firing rate of ChIs was increased without an increase in burstiness, closely mimicking changes observed in the ChIs from L-DOPA treated mice. Likewise, partial blockade of both HCN and SK channels caused significant depression of ChI firing rate, consistent with the DA depleted state. These findings confirm that reducing the current from these channels is sufficient to account for the changes in ChI activity and show that the slower rate of depolarization from hyperpolarized potentials caused by HCN inhibition overrides the smaller AHP resulting from reduced SK activity. Importantly, further studies should address whether targeted restoration of these channels' activities can be employed to ameliorate PD-related motor deficiencies, such as akinesia and LID.

In summary, we followed up on our previous finding that chronic L-DOPA treatment of DA lesioned mice is associated with increased ChI firing and now demonstrate a novel cellular mechanism that implicates changes in HCN and SK currents that results from DA depletion and chronic L-DOPA treatment. Critically, these changes are evident beyond the pharmacological time course of L-DOPA, suggesting that they reflect the physiological state of ChIs upon which DA replacement works in patients treated with L-DOPA. Interestingly, we found that only some of the changes in ChI physiology caused by DA depletion are restored by chronic L-DOPA treatment and some persisted despite treatment. Our data suggest that HCN and SK channels can be targeted for therapeutic intervention, although development of cell type-specific modulators of channel activities is desired for effective and safe behavioral outcomes.

# Materials and methods

**Key resources table**

| Reagent type (species) or resource | Designation | Source or reference | Identifiers | Additional information |
|---|---|---|---|---|
| Strain, strain background (*Mus musculus* MF) | C57BL/6J mouse | Jackson Laboratory | IMSR Cat# JAX:000664, RRID:IMSR_JAX:000664 | |
| Genetic reagent (*Mus musculus* MF) | *Chat*-Cre; Tg(*Chat*-cre)GM24Gsat/Mmucd | GENSAT | RRID:MMRRC_017269-UCD | Colony maintained in-house |
| Genetic reagent (*Mus musculus* MF) | Ribotag; B6N.129-*Rpl22*<sup>tm1.1Psam</sup>/J | Jackson Laboratory | IMSR Cat# JAX:011029, RRID:IMSR_JAX:011029 | Colony maintained in-house |
| Chemical compound, drug | 6-OHDA ; 6-hydroxydopamine hydrobromide | Sigma Aldrich | Cat#:H-116 | |
| Chemical compound, drug | Desipramine hydrochloride | Sigma Aldrich | Cat#:D-3900 | |
| Chemical compound, drug | L-DOPA; L-3,4-dihydroxyphenylalanine methyl ester hydrochloride | Sigma Aldrich | Cat#:D-1507 | |

*Continued on next page*

*Continued*

| Reagent type (species) or resource | Designation | Source or reference | Identifiers | Additional information |
|---|---|---|---|---|
| Chemical compound, drug | Benserazide hydrochloride | Sigma Aldrich | Cat#:D-7283 | |
| Chemical compound, drug | TTX; tetrodotoxin citrate | Tocris | Cat#:1069 | |
| Chemical compound, drug | ZD7288 | Tocris | Cat#:1000 | |
| Chemical compound, drug | apamin | Tocris | Cat#:1652 | |
| Chemical compound, drug | CNQX; CNQX disodium salt | Tocris | Cat#:1045 | |
| Chemical compound, drug | AP5; DL-AP5 | Tocris | Cat#:0105 | |
| Chemical compound, drug | PCX; picrotoxin | Tocris | Cat#:1128 | |
| Chemical compound, drug | QX 314 bromide | Tocris | Cat#:1014 | |
| Chemical compound, drug | biocytin | Sigma-Aldrich | Cat#:B4261 | |
| Chemical compound, drug | dopamine; dopamine hydrochloride | Sigma-Aldrich | Cat#:H8502 | |
| Chemical compound, drug | $BaCl_2$ | Sigma-Aldrich | Cat#:449644 | |
| Chemical compound, drug | gramicidin | Sigma-Aldrich | Cat#:G5002 | |
| Chemical compound, drug | streptavidin-DyLight-633 conjugate | Thermo Scientific | Cat:21844 | |
| Software, algorithm | WinWCP | University of Strathclyde, UK | RRID:SCR_014713 | http://spider.science.strath.ac.uk/sipbs/software_ses.htm |
| Software, algorithm | Pclamp10 | Molecular Devices | RRID:SCR_011323 | |
| Software, algorithm | Igor Pro 6 | WaveMetrics | RRID:SCR_000325 | |
| Software, algorithm | Mini Analysis | Synaptosoft | RRID:SCR_002184 | |
| Software, algorithm | Matlab 2016a, Matlab 2019a | MathWorks | RRID:SCR_001622 | |
| Software, algorithm | GraphPad Prism v8 | GraphPad | RRID:SCR_002798 | |

## Animals

The use of the animals followed the National Institutes of Health guidelines and was approved by the Institutional Animal Care and Use Committee of Columbia University and New York State Psychiatric Institute. For behavior and slice electrophysiology studies, we used male C57BL/6J mice (Jackson Laboratory, Bar Harbor, ME, stock# 000664) at twelve-weeks of age at the beginning of the experiments. To obtain bitransgenic mice that express 'tagged' ribosomes selectively in cholinergic neurons, mice expressing the Cre-recombinase under the regulation of the *Chat* promotor (*Chat-Cre*; Tg(*Chat-Cre*)GM24sat/Mmucd from the GENSAT Project obtained from the MMRRC, stock# 017269-UCD) (*Gong et al., 2007*) were bred with mice expressing a Cre-activated knock-in of HA-tagged *Rpl22* ribosomal subunit (ribotag mice; B6N.129-*Rpl22*tm1.1Psam/J obtained from Jackson Laboratory, stock# 011029) (*Sanz et al., 2009*). Mice of both sexes were used for experiments and were homozygous for ribotag and heterozygous for *Chat-Cre*.

## DA lesion and chronic L-DOPA treatments

Anesthesia was induced by intraperitoneal (IP) injection of ketamine and xylazine, followed by a subcutaneous injection of bupivacaine for local anesthesia at the incision site. Animal were head-fixed in a stereotaxic apparatus (Kopf Instruments) with ear cups, and 6-hydroxydopamine (6-OHDA, Sigma, St. Louis, MO; H-116; 4.5 ug dissolved in 1.5 microliters of 0.05% ascorbic acid in 0.9% saline) was injected into left medial forebrain bundle (MFB) (coordinates: AP −1.3 mm and ML +1.3 mm from Bregma, and DV −5.4 mm from skull surface) through a small borehole in the skull. The entire volume was infused over 7.5 min through a stainless-steel cannula (Braintree Scientific, Braintree, MA; RM-SBL STD), which was left in place for an additional 5 min before withdrawal and incision closure. Desipramine (Sigma D3900; 25 mg/kg delivered IP) was given 30 min prior to 6-OHDA infusion to block uptake of the toxin by noradrenergic neurons. Intensive post-operative care included providing supplemental nutrition (Bacon Softies F-3580, Bio-Serv, Flemmington, NJ) and extra fluids (saline subcutaneously and dextrose saline IP). The health status of the animals was monitored daily until stabilization of body weight with free access to food and water. Sham-lesioned control mice received same volume of vehicle into the left MFB.

3–4 weeks after unilateral 6-OHDA injection, lesioned mice were tested for stepping (*Figure 1—figure supplement 1A*) and randomly divided to receive daily IP injections of saline or L-DOPA (3 mg/kg + 12.5 mg/kg benserazide), while all sham-injected control mice received saline. For slice electrophysiology studies, mice were used at 3–11 weeks after the first injection of saline or L-DOPA. For gene expression analysis, striatal tissue was collected after 3 weeks of daily L-DOPA treatment. Dopaminergic lesion was confirmed in some cohorts of mice by western blot analysis of striatal tissue lysates for tyrosine hydroxylase (TH) protein levels which showed near >98% depletion of TH in the striatal hemisphere ipsilateral to the 6-OHDA lesion (not shown).

## Weight-supported treadmill stepping task

Stepping tests to assess akinesia of the impaired forelimb (contralateral to the 6-OHDA lesion) were performed 3–4 weeks post 6-OHDA injection before the initiation of repeated L-DOPA or saline treatment. Both forelimbs were placed on a treadmill with the surface moving at 4.6 cm/sec away from the head of the mouse while the body weight was supported by examiner. Forepaw steps were video recorded from five nonconsecutive cycles of treadmill. The number of left and right paw steps was counted over a distance of 45 cm per trial for five non-consecutive trials and averaged to obtain the stepping score per mouse.

## LID assessment

L-DOPA-induced abnormal involuntary movements (AIMs) were assessed after the first IP injection of L-DOPA (the acute L-DOPA in naïve state) and again after 3 weeks of daily L-DOPA treatment (chronically-treated state) (*Figure 1—figure supplement 1B–D*). One cohort of mice was tested again at 10 weeks of L-DOPA treatment, which showed no difference from the 3 week time point indicating that LID scores remained stable throughout the time course of the experiments (not shown). At the start of the session, each mouse was placed into a clear polypropylene cylinder and allowed to acclimate for at least 3 min. L-DOPA was then injected and the mouse was video-recorded for 1 min periods at 0, 5, 10, 20, 40, 60, 80, 100 and 120 min post injection. Limb and axial

dyskinesias were analyzed from recorded videos using previously described protocols (*Ding et al., 2011*; *Won et al., 2014*).

## Slice preparation and electrophysiological recordings

At the time of recordings, mice were 4–6 month-old; each day, animals were randomly selected from a different treatment group. Slices were prepared at least 20 hr after the last L-DOPA or vehicle treatment. Mice were euthanized by cervical dislocation and coronal 270 µm-thick striatal slices were prepared on a vibratome (VT1200; Leica, Sloms, Germany) in oxygenated ice cold cutting-artificial cerebrospinal fluid (ACSF) containing (in mM): 194 sucrose, 30 NaCl, 4.5 KCl, 26 NaHCO$_3$, 6 MgCl$_2$·6H$_2$O, 1.2 NaH$_2$PO$_4$, and 10 D-glucose (pH 7.4, 290 ± 5 mOsm). Slices were then transferred to oxygenated normal ACSF containing 125.2 NaCl, 2.5 KCl, 26 NaHCO$_3$, 1.3 MgCl$_2$·6H$_2$O, 2.4 CaCl$_2$, 0.3 NaH$_2$PO$_4$, 0.3 KH$_2$PO$_4$, and 10 D-glucose (pH 7.4, 290 ± 5 mOsm) at 34°C and allowed to recover for at least 40 min before the recordings.

Electrophysiological recordings were performed on an upright Olympus BX50WI (Olympus, Tokyo, Japan) microscope equipped with a 40x water immersion objective, differential interference contrast (DIC) optics and an infrared video camera. All recorded ChIs were located in the dorsolateral striatum and identified by larger somatic size than neighboring neurons. Slices were transferred to a recording chamber and maintained under perfusion with normal ACSF (1.5–2 mL/min) at 34°C. Patch pipettes (3–5 MΩ) were pulled using P-97 puller (Sutter instruments, Novato, CA) and filled with internal solutions as indicated below. All chemicals for ACSF as well as gramicidin, biocytin, dopamine, picrotoxin, sulpiride, and SCH23390 were purchased from Sigma. TTX, apamin, ZD7288, APV, and CNQX were from Tocris (Bristol, UK). Patch clamp recordings were performed with a Multi-Clamp 700B amplifier (Molecular Devices, Forster City, CA) and digitized at 10 kHz with InstruTECH ITC-18 (HEKA, Holliston, MA). Data were acquired using WINWCP software (developed by John Dempster, University of Strathclyde, UK) and analyzed using Clampfit (Molecular Devices), Igor Pro (Wavemetrics, Lake Oswego, OR), and Matlab (MathWorks, Natick, MA).

The pipette solution contained (in mM): 115 K-gluconate, 10 HEPES, 2 MgCl$_2$, 20 KCl, 2 MgATP, 1 Na$_2$-ATP, and 0.3 GTP (pH = 7.3; 280 ± 5 mOsm). After recording cell firing rate in cell-attached mode, rheobase was measured by first applying a hyperpolarizing −10 pA/sec ramp for 5 s, then applying a depolarizing 55 pA/sec current ramp to +500 pA, followed by a −15 pA/sec ramp to 0 pA (*Figure 1—figure supplement 3E*; *Maurice et al., 2004*). Next, the neuronal membrane was ruptured and basal electrophysiological characteristics of ChIs, including IV curve, RMP, input resistance and membrane capacitance were measured in current-clamp mode. For perforated-patch recordings, 60 mg/ml gramicidin was added to the patch pipette solution.

Glutamatergic spontaneous excitatory postsynaptic currents (sEPSCs) were recorded in whole-cell voltage clamp mode in cells pre-treated with 25 µM picrotoxin to inhibit GABA$_A$ receptors; the internal pipette solution contained (in mM) 120 CsMeSO$_3$, 5 NaCl, 10 HEPES, 1.1 EGTA, 2 Mg$^{2+}$-ATP, 0.3 Na-GTP, 2 Na-ATP, and 5 QX314 to block of voltage-activated Na$^+$ channels (pH = 7.3, 280 ± 5 mOsm). GABAergic spontaneous inhibitory postsynaptic currents (sIPSCs) were recorded in the presence of 25 µM APV, and 10 µM CNQX (or NBQX) to inhibit glutamatergic receptors; internal pipette solution contained (in mM): 140 CsCl, 2 MgCl$_2$, 10 HEPES, 2 EGTA, 2 MgATP, 1 Na$_2$-ATP, 0.3 GTP and 5 QX314 (pH = 7.3, 280 ± 5 mOsm). Postsynaptic currents were detected and analyzed using Mini Analysis program (Synaptosoft, Decatur, GA). The threshold for amplitude detection was 8–10 pA which was >2 fold the RMS of the background noise.

Voltage sag was measured in current-clamp mode following 500ms-long current injections from 0 to −300 pA. To measure $I_h$ currents in voltage-clamp mode in the presence of TTX, ChIs were held at −60 mV, then depolarized to −40 mV followed by 1 s hyperpolarizing steps to −140 mV in 10 mV voltage increments. Sensitivity to 25 µM ZD7288 was used to confirm that the voltage sag and $I_h$ were mediated by HCN channels. Average access resistance in V-clamp recordings was 12.3 ± 5.4 MΩ; recordings with access resistance of more than 30 MΩ or those where it changed by more than 25% after the application of ZD7288 were discarded (3 of 14 cell in sham; 2 of 14 cells in 6-OHDA; 5 of 18 cells in chronic LD in *Figure 3F,G*). Similarly, recordings with input resistance of more than 600 MΩ (mean+2.5SD; *Figure 1—figure supplement 3C*) in I-clamp mode were discarded (7 of 46 cell in sham; 2 of 25 cells in 6-OHDA; 1 of 31 cells in chronic LD in *Figure 3D*).

To measure Ba$^{2+}$-sensitive afterhyperpolarization (AHP) currents, cells were clamped at −60 mV in the presence of TTX followed by a 300ms-long depolarizing step to +10 mV. *Peak* current

amplitude was measured in the first 200 ms following the offset of the depolarizing voltage step, while the *late* phase was the average steady-state current at 900–1000 ms after the end of the step. To measure apamin-sensitive current, cells were depolarized from −60 mV to 0 mV for 100 ms. $BaCl_2$ (200 μM) or apamin (0.5–100 nM) were applied for 10 min and currents measured after drug application were subtracted from those before the treatments to assess the contribution of *peak* and *late* components of AHP.

## Analysis of Burst-Pause activity

We used the Robust Gaussian Surprise (RGS) method to determine burst and pause patterns during tonic ChI firing using the MatLab code available from *Storey et al., 2016*. This method identifies differences in the firing rate of adjacent spikes against the local log ISI distributions, assigning individual spikes to burst or pause strings based on statistical criteria in comparison to the Gaussian distribution of the entire spike train (*Ko et al., 2012*). The parameters used for burst and pause detection were: p=0.15 for the calculation of the central location, alpha = 0.05 for Bonferroni correction, $N_{min}$ = 2 for minimum number of spikes to be considered a burst/pause, and central distribution calculated as the median ±2 x median average deviation.

## Immunohistochemistry and morphological analysis

For morphological characterization of recorded ChIs, biocytin (1 mg/ml) was included in the internal pipette solution and allowed to fill the cell for 30–40 min after achieving the whole-cell configuration. Then, slices were fixed with 4% paraformaldehyde in 0.1M PBS overnight at 4℃, washed with Tris-buffered saline (TBS: 50 mM Tris-Cl, 150 mM NaCl, pH 7.5) and incubated with a streptavidin-Dylight 633 conjugate (1:200; Thermo Scientific, Grand Island, NY) in TBS + 0.6% Triton-X100 for 48 hr. The slices were then rinsed and mounted on glass slides using Fluormount-G (Southern Biotech, Birmingham, AL). For morphological reconstruction, serial optical sections encompassing the neurites of biocytin-labeled ChIs were imaged at 0.25 $μm^2$ pixels at a z-depth of 0.74 μm using a Leica DM6 confocal microscope with a 20x/0.7 NA oil immersion objective (Leica HCX PL APO CS). The neurites were traced from the resulting stacks using the Simple Neurite Tracer plugin in Fiji (ImageJ) for Sholl analysis (*Ferreira et al., 2014*; *Longair et al., 2011*).

## Gene expression analysis

Brain tissue from *Chat-Cre* x *Rpl22*[HA] mice was collected at least 20 hr after the last L-DOPA or saline injection. Striatal tissue ipsilateral to the 6-OHDA lesion from 3 to 4 mice for each experimental group was pooled per replicate (4–6 total replicates per condition), weighed and immediately placed into cold homogenization buffer at 5% (w/v) consisting of (pH 7.4, in mM): 50 Tris, 100 KCl, 12 $MgCl_2$, 1 DTT, 1% Nonidet P40 substitute (Roche, Basel, Switzerland), 0.1 mg/ml cyclohexamide, 1x protease inhibitor cocktail, and 200 U/mL RNAsin (Promega, Madison, WI). The tissue was then homogenized with a powered Dounce homogenizer at 1700 rpm for 14 complete up-down strokes, followed by centrifugation at 10,000 RCF for 10 min at 4℃. Mouse anti-HA antibody (5 μl, HA.11, BioLegend, San Diego, CA; MMS-101R) was added to 800 μl of the resulting supernatant and incubated for 4 hr at 4℃. The mixture was then added to protein-G magnetic beads (Dynabeads 400 μl equivalent, Invitrogen, Carlsbad, CA) and incubated overnight at 4℃. The next day, the beads were separated from the supernatant with a magnet and washed 3 times for 10 min at 4℃ in high salt washing buffer consisting of (in mM): 50 Tris, 300 KCl, 12 $MgCl_2$, 0.5 DTT, 1% Nonidet P40 substitute, and 0.1 mg/ml cyclohexamide. After the last wash, the beads were collected and the bound RNA eluted with 350 μl of RLT buffer from the RNeasy Micro Kit. The beads were removed from the RLT buffer and the RNA was isolated according the manufacturer's instructions (Qiagen, Hilden, Germany). The resulting RNA was assayed for integrity and amount with a Bioanalyzer (Agilent, Santa Clara, CA). All samples exhibited a RNA integrity number (RIN) score of 8.4–10.

cDNA libraries were generated from the resulting RNA with the SuperScript IV VILO master mix (Invitrogen) according to the manufacturer's protocols. Gene expression was measured using Taqman probes for the target genes and *Actb* as a housekeeping control assayed in duplex for each well on a CFX96 Touch Real-Time PCR Detection System (Bio-Rad, Hercules, CA) using TaqMan Fast Advanced (Applied Biosystems, Foster City, CA) master mix according to the manufacturer's instructions. Each sample was assayed in triplicate using 0.33 μl of the cDNA library (undiluted) per

reaction. Cycling conditions were: 50° x 2 mins, 95° x 20 s, (95° x 3 s, 60° x 30 s) x 40 cycles. Gene expression is stated as a ratio of the target gene to *Actb* as determined by $2^{-\Delta Ct}$. The probes were obtained from Applied Biosystems and include: *Hcn1* (Mm00468832_m1), *Hcn2* (Mm00468538_m1), *Hcn3* (Mm01212852_m1), *Hcn4* (Mm01176086_m1), *Pex5l* (Mm00458088_m1), *Chat* (Mm01221882_m1), *Kcnn1* (Mm01349167_m1), *Kcnn2* (Mm00446514_m1), *Kcnn3* (Mm00446516_m1) and *Actb* (4352341E).

## Statistical analysis

Unless stated otherwise, electrophysiological data represent observations from single neurons in slices. We consider each neuron to be a biological replicate, with no technical replicates included in any statistical analysis. The total number of neurons and animals used for each experiment are indicated in the text, figure legends and source data files for each figure. Sample sizes were calculated using G*Power software for three experimental groups assuming the typical variance from our data, effect size = 0.3, $\alpha$-error = 0.05, and power = 0.95 for physiology experiments and indicated at least 7 data points were needed for each group. For the mRNA expression data, the number of animals per replicate pool were selected based on *Doyle et al., 2008* where polyribosomal immunoprecipitation was used to isolate cell-type specific mRNA from striatal tissue. Mice were randomly assigned to receive sham or 6-OHDA lesions. The lesioned animals were again randomized to receive chronic treatment with saline of L-DOPA. The treatment groups were not masked for allocation, data collection, or analysis. For data expressed as dot plots, each symbol represents an individual neuron, the horizontal line denotes the median, and whiskers the interquartile range (25-75th percentile). For bar and line graphs, data are expressed as mean ± standard error of the mean (SEM). Data in the text are expressed as mean ± SEM. The statistical tests used are listed in the figure legends and text. Unless indicated, non-parametric tests were used for all figures where possible. All individual cell data and details of statistical tests and data representation are included in the source data files for each figure. Statistical analysis and data were plotted using GraphPad Prism 8.4 (GraphPad Software, San Diego, CA).

## Acknowledgements

This work was supported by NINDS R01NS101982 (UJK), NINDS R01NS075222 (EVM), the JBP Foundation (DS) and NIDA R0107418 (DS). We thank Nicolas Tritsch, Ori Lieberman and Avery McGuirt for helpful comments and Joanna Garcia for technical assistance.

## Additional information

### Funding

| Funder | Grant reference number | Author |
| --- | --- | --- |
| National Institute of Neurological Disorders and Stroke | R01NS101982 | Un Jung Kang |
| National Institute of Neurological Disorders and Stroke | R01NS075222 | Eugene V Mosharov |
| National Institute of Neurological Disorders and Stroke | R01DA007418 | David Sulzer |
| The JBP foundation | | David Sulzer |

The funders had no role in study design, data collection and interpretation, or the decision to submit the work for publication.

### Author contributions

Se Joon Choi, Data curation, Formal analysis, Investigation, Visualization, Methodology, Writing - review and editing; Thong C Ma, Data curation, Formal analysis, Validation, Investigation, Visualization, Methodology, Writing - original draft, Writing - review and editing; Yunmin Ding, Conceptualization, Data curation, Formal analysis, Investigation, Visualization, Methodology, Writing - review

and editing; Timothy Cheung, Formal analysis, Investigation, Methodology, Writing - review and editing; Neal Joshi, Data curation, Methodology; David Sulzer, Supervision, Writing - review and editing; Eugene V Mosharov, Conceptualization, Data curation, Formal analysis, Supervision, Investigation, Visualization, Writing - original draft, Project administration, Writing - review and editing; Un Jung Kang, Conceptualization, Resources, Supervision, Funding acquisition, Investigation, Writing - original draft, Project administration, Writing - review and editing

### Author ORCIDs
Thong C Ma ⓘ http://orcid.org/0000-0002-9395-0448
Un Jung Kang ⓘ https://orcid.org/0000-0002-5970-6839

### Ethics
Animal experimentation: This study was performed in strict accordance with the recommendations in the Guide for the Care and Use of Laboratory Animals of the National Institutes of Health. All of the animals were handled according to approved institutional animal care and use committee (IACUC) protocols (#08-133) of the University of Arizona. The protocol was approved by the Committee on the Ethics of Animal Experiments of the University of Minnesota (Permit Number: 27-2956). All surgery was performed under sodium pentobarbital anesthesia, and every effort was made to minimize suffering.

### Decision letter and Author response
Decision letter https://doi.org/10.7554/eLife.56920.sa1
Author response https://doi.org/10.7554/eLife.56920.sa2

## Additional files
### Supplementary files
• Transparent reporting form

### Data availability
All data generated for analyzed during this study are included in the manuscript and supporting files. Source data file has been provided for Figures 1-5 and Figure 1—figure supplements 1–3.

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
