## [Decision Letter]

**Acceptance summary:**

Prior work showed that striatal cholinergic interneurons are required for levodopa-induced dyskinesia, but it was unclear whether and how cholinergic interneuron properties changed in these conditions. Using ex vivo electrophysiology, this paper provides convincing evidence that cholinergic interneurons indeed show changes in their spontaneous firing rates, and furthermore identified that both HCN and SK ionic currents are involved in the underlying cellular mechanisms. This cellular mechanism of both translational and basic science interest, and provides important insights into the understanding of levodopa-induced dyskinesia.

**Decision letter after peer review:**

Thank you for submitting your article "Alterations in the intrinsic properties of striatal cholinergic interneurons after dopamine lesion and chronic L-DOPA" for consideration by *eLife*. Your article has been reviewed by three peer reviewers, including Jun Ding as the Reviewing Editor and Reviewer #1, and the evaluation has been overseen by John Huguenard as the Senior Editor. The following individual involved in review of your submission has agreed to reveal their identity: Joshua A Goldberg (Reviewer #3).

The reviewers have discussed the reviews with one another and the Reviewing Editor has drafted this decision to help you prepare a revised submission.

The reviewers are impressed by the thorough analysis of the properties of striatal cholinergic interneurons (ChI) in parkinsonian animals chronically treated with levodopa. The topic is of both translational and basic science interest. In addition, the reviewers think that this will be one of the highest quality studies in ChI in the literature, once published. The physiology is of a very high quality, the writing is exceptionally clear, and the overall conclusions are well-justified.

The reviewers did raise some concerns. However, very minor additional experiments have been requested. The main revision should focus on clarity and presentation, and critically on data reporting.

1) Data reporting: please report both number of cells and number of animals in each figure.

2) Data reporting: for readability, it would be useful to put numbers for some of the major findings in the Results text, not only in the figures/figure legends, mean =/- SEM. For example, the average firing rate of ChI and the n for the experiment.

3) The experiment described in Supplementary Figure 3 is a bit confusing. Please clarify and rationale and conclusion from the D1 and D2 antagonist experiments.

4) In some data sets, the number of replicates is small, for example, n=3-8. Even though the conclusion is clear, the reviewers think that the minimum n number for such electrophysiology experiments should be n=6-8.

5) Please add some discussion of whether L-DOPA treatment alone (in healthy animals) could potentially affect ChI physiology.

6) Please add discussion on the potential relationship between change of ChI dendritic morphology and physiological findings.

Reviewer #1:

It has been shown for decades that cholinergic modulation mediated by striatal cholinergic interneuron (ChI) and activity of ChI itself are involved in Parkinson's disease (PD) and L-dopa induced dyskinesia (LID). However, the basis of changes in ChI physiology in PD and LID is still not clear. In the manuscript by Choi and colleagues, the authors performed thorough electrophysiological analysis on the firing of ChI in acute brain slices from control, DA depleted and L-dopa treated mice. They identified that both HCN and SK currents were decreased in the DA depleted condition, in addition, HCN currents were rescued after L-dopa treatment, but SK current remained depressed after chronic L-dopa. These changes of intrinsic properties can explain the decreased firing frequency seen in DA depleted state, and elevated firing frequency in L-dopa treated condition. To confirm the changes in HCN and SK, the authors performed thorough pharmacological manipulations and analysis, as well as the HCN and SK mRNA expression levels. It would be fantastic to directly manipulate ion channel activity selectively in ChIs in vivo and examine the behavioral consequences. However, such manipulation (especially fine adjustment of ion channel activity to certain level in vivo) is technically not possible. Overall, the data quality is high. It would be one of the most thorough analysis of ChI physiology in DA depletion and L-dopa treatment conditions. The overall conclusion is well supported by the experimental data. I do have a couple of concerns/suggestions:

1) What is the effect of chronic L-dopa treatment on ChI physiology? This would be valuable to access the increased firing is caused by L-dopa treatment or requires prior DA depletion.

2) What is the relationship between the change of ChI dendritic arborization and HCN, and/or SK current? This is particularly relevant because HCN is highly expressed in dendrites. More discussion would be helpful.

3) The statistical and data reporting can be further improved.

4) The data included in Supplementary Figure 3 are particularly confusing. It is not clear what the authors want to tell here. It seems that either D1 or D2 antagonist alone could significantly decrease the firing frequency in DA depleted and LID conditions. However, the statistical analysis here is very confusing.

Reviewer #2:

In this manuscript, Choi et al. examine the properties of striatal cholinergic interneurons (ChI) in parkinsonian animals chronically treated with levodopa. Prior work showed that ChI are required for levodopa-induced dyskinesia (LID), but it was unclear whether and how ChI properties changed in these conditions. Using ex vivo electrophysiology, the authors find that ChI indeed show changes in their spontaneous firing rates, and furthermore are able to identify some of the underlying cellular mechanisms. The topic is of both translational and basic science interest, and this will be one of the highest quality studies in ChI in the literature, once published. The physiology is of a very high quality, the writing is exceptionally clear, and the overall conclusions are well-justified. However, I have a few concerns:

1) The authors have three main conditions: healthy control, 6-OHDA (dopamine depleted), and 6-OHDA (dopamine depleted) + levodopa treated. Their key finding is that the firing rate of ChI is reduced in the 6-OHDA condition, but restored to above healthy control levels in the 6-OHDA + levodopa condition. Ideally, they would have included a fourth group, healthy control + levodopa, to assay whether levodopa treatment can cause changes in healthy ChI as well, or whether there is a synergy between the parkinsonian striatal changes and the levodopa. If they are unable to include/add this fourth group, it would be valuable to comment on what the authors think might be going on in this condition.

2) The critical findings of this paper are related to the intrinsic properties of ChI, and the numbers of cells recorded are quite large for these primary analyses. However, they have also included data related to ChI synapses (sIPSC, sEPSCs), which have much smaller sample sizes, and indeed, one of their experiments (sEPSCs) shows no difference between conditions. Given the typical variability in sEPSC and sIPSC or mEPSC and mIPSC measurements, from slice to slice, mouse to mouse, and cell to cell, I think a larger n is required to definitively say whether ChI synapses are similar or different across conditions. I would favor increasing the n, dropping this section, or rephrasing the text to acknowledge these experiments are underpowered.

3) At the end of the Results section, the authors include data where they have washed on dopamine. While potentially interesting, the relationship of these experiments to their overall result is not clearly articulated, and again, the n is exceptionally small (3-8) by comparison to the robust numbers for their central findings.

Reviewer #3:

The manuscript of Choi and co-workers addresses the issue of alterations in the firing rate of striatal cholinergic interneurons (ChIs) upon treatment with 6-OHDA, which causes hemiparkinsonism, and then following chronic levodopa induced dyskinesia. While there is clear evidence for decades that dopamine depletion leads to adaptation in cholinergic signaling in the striatum, it has been difficult to determine what actually happens to ChIs which are the main source of tonic striatal acetylcholine (ACh). Several papers have addressed this issue and have reach contradictory results. I myself have tried this for decades while at various labs including my own and have not been able to reach a clear finding. Dr. Kang is one of the first to have reported a clear increase in the firing rate of ChIs after induction of LID in a seminal paper published a decade ago. This finding has been replicated since by several groups, but whether there is a change after 6-OHDA treatment has been hard to tell.

In this manuscript, the authors put this question to rest with incontrovertible evidence that there is indeed a reduction in the firing rate of ChIs. But more importantly, the authors have provided compelling mechanistic evidence that the HCN and SK currents are responsible for these changes in firing rate. They have done this with the use of direct measurements of these currents showing that they are altered with the various treatments. They also showed that pharmacological manipulation of these currents can phenocopy the effect. It would have been nice to have seen a rescue experiment, however, this is not really possible because we don't have reliable drugs to boost HCN or SK currents. They have also measured the transcript of these channels subunits. So, as far as I'm concerned, they have done an excellent job.

In addition to the high technical quality and commendable rigor of the work, and the resolution of this long standing question, there is great clinical relevance to this finding. It has been known for decades that the anticholinergic therapy can be effective to treat parkinsonism, albeit with unpleasant side effects. Discovering the precise biophysical mechanism of the adaptation in ChI firing rates, provides new potential therapeutic targets. For these reasons, I find this manuscript, highly relevant to the general audience of neuroscientists and biologists.

I think this study is well done and basically ready.

---

## [Author Response]

The reviewers did raise some concerns. However, very minor additional experiments have been requested. The main revision should focus on clarity and presentation, and critically on data reporting.1) Data reporting: please report both number of cells and number of animals in each figure.

Both the number of cells and animals are now reported in the text and figure legends, as well as the source data files for each figure containing all data sets and analysis. In addition, we have replotted our physiology data using dot plots to show all data points and have incorporated an additional dataset that was analyzed after the initial submission into Figure 3D.

2) Data reporting: for readability, it would be useful to put numbers for some of the major findings in the Results text, not only in the figures/figure legends, mean =/- SEM. For example, the average firing rate of ChI and the n for the experiment.

We now report the values of our major findings in the text as mean ± SEM along with statistical tests used for analysis.

3) The experiment described in Supplementary Figure 3 is a bit confusing. Please clarify and rationale and conclusion from the D1 and D2 antagonist experiments.

The original idea was to show that dopamine depletion and chronic L-DOPA treatment altered ChI response to dopamine. For this, we used a combination of D1 and D2 antagonists to determine relative contributions of D1- and D2-like receptors in mediating the response to dopamine. As these experiments as a whole are relatively underpowered (n=3-8) and we don’t currently have a means to increase the N, we have decided to remove these data from the manuscript. The figures showing the response to dopamine (n=7-10) are now presented on Figure 1F and G.

4) In some data sets, the number of replicates is small, for example, n=3-8. Even though the conclusion is clear, the reviewers think that the minimum n number for such electrophysiology experiments should be n=6-8.

We have removed the experiments with low replicates (old Supplementary Figure 3C and D). All of our data reported for electrophysiology experiments are from at least 6 replicates.

5) Please add some discussion of whether L-DOPA treatment alone (in healthy animals) could potentially affect ChI physiology.

In our previous publication, we used *aphakia* mice, which lack dopaminergic projections to the dorsal striatum due to a mutation in *Pitx3*. Only mice harboring homozygous *Pitx3* mutations exhibit LID and show increased ChI firing rate and responsiveness to dopamine with chronic L-DOPA treatment. Heterozygous *Pitx3* mutant mice, which are phenotypically and dopamine normal, do not exhibit LID or changes in these parameters of ChI physiology when treated with L-DOPA in the same way (Ding et al., 2007). These findings reinforce that dopamine depletion is needed to bring about chronic L-DOPA-mediated changes to ChI physiology. We have addressed this in the Discussion.

6) Please add discussion on the potential relationship between change of ChI dendritic morphology and physiological findings.

We have added discussion regarding how localization (or lack thereof) of HCN channels to newly formed dendrites may affect excitability in response to synaptic input as with cortical and hippocampal pyramidal neurons. As we found a decrease in Trip8b mRNA with dopamine depletion, HCN channels may not be appropriately trafficked into dendrites, suggesting that this may cause increased dendritic excitability, despite decreased spontaneous activity. Additionally, we discuss a role for dendritic SK2 vs. somatic SK3 channels in regulating firing pattern vs. rate as seen in dopaminergic neurons. These points are added in the Discussion.

Reviewer #1:[…] 1) What is the effect of chronic L-dopa treatment on ChI physiology? This would be valuable to access the increased firing is caused by L-dopa treatment or requires prior DA depletion.

Treating dopamine intact mice chronically with L-DOPA did not change ChI physiology in our previous study. Please see Reviewing Editor comment #5 above.

2) What is the relationship between the change of ChI dendritic arborization and HCN, and/or SK current? This is particularly relevant because HCN is highly expressed in dendrites. More discussion would be helpful.

We have added discussion regarding the regulation of excitability by the dendritic localization of HCN channels in other neuron types and the possible role of somatic vs. dendritic SK channels. Please see Reviewing Editor comment #6 above.

3) The statistical and data reporting can be further improved.

We now report the mean ± SEM of our observations in the text, and complete information on the number of neurons and mice in the text and figure legends. Please see Reviewing Editor comments #1 and #2 above.

4) The data included in Supplementary Figure 3 are particularly confusing. It is not clear what the authors want to tell here. It seems that either D1 or D2 antagonist alone could significantly decrease the firing frequency in DA depleted and LID conditions. However, the statistical analysis here is very confusing.

We have removed the experiments using D1 and D2 antagonists due to low number of replicates and have moved the figures showing response to dopamine to Figure 1F and G. Please see Reviewing Editor comment #3.

Reviewer #2:[…] 1) The authors have three main conditions: healthy control, 6-OHDA (dopamine depleted), and 6-OHDA (dopamine depleted) + levodopa treated. Their key finding is that the firing rate of ChI is reduced in the 6-OHDA condition, but restored to above healthy control levels in the 6-OHDA + levodopa condition. Ideally, they would have included a fourth group, healthy control + levodopa, to assay whether levodopa treatment can cause changes in healthy ChI as well, or whether there is a synergy between the parkinsonian striatal changes and the levodopa. If they are unable to include/add this fourth group, it would be valuable to comment on what the authors think might be going on in this condition.

Treating dopamine intact mice chronically with L-DOPA did not change ChI physiology in our previous study. Please see Reviewing Editor comment #5 above.

2) The critical findings of this paper are related to the intrinsic properties of ChI, and the numbers of cells recorded are quite large for these primary analyses. However, they have also included data related to ChI synapses (sIPSC, sEPSCs), which have much smaller sample sizes, and indeed, one of their experiments (sEPSCs) shows no difference between conditions. Given the typical variability in sEPSC and sIPSC or mEPSC and mIPSC measurements, from slice to slice, mouse to mouse, and cell to cell, I think a larger n is required to definitively say whether ChI synapses are similar or different across conditions. I would favor increasing the n, dropping this section, or rephrasing the text to acknowledge these experiments are underpowered.

We agree that a larger number of observations would be desirable for these experiments, especially due to the low frequency of sEPSCs in ChI. As such, we added “However, due to the rarity of spontaneous excitatory events, we are likely underpowered for a definite conclusion about the lack of change in sEPSCs,” to the Results.

3) At the end of the Results section, the authors include data where they have washed on dopamine. While potentially interesting, the relationship of these experiments to their overall result is not clearly articulated, and again, the n is exceptionally small (3-8) by comparison to the robust numbers for their central findings.

These experiments were designed to show that dopamine depletion and chronic L-DOPA treatment alters response to dopamine. We agree that the number of replicates for experiments using D1 and D2 antagonist needs to be increased and have decided to remove these from the manuscript. The remaining data has been moved to Figure 1. Please see Reviewing Editor comment #3 and #4 above.

Reference:

Ding, Y., Restrepo, J., Won, L., Hwang, D.Y., Kim, K.S., and Kang, U.J. (2007). Chronic 3,4-dihydroxyphenylalanine treatment induces dyskinesia in aphakia mice, a novel genetic model of Parkinson's disease. Neurobiol Dis 27, 11-23.